# Statistical atmospheric inversion of local gas emissions by coupling the tracer release technique and local scale transport modelling: a test case with controlled methane emissions

Sébastien Ars[1], Grégoire Broquet[1], Camille Yver Kwok[1], Yelva Roustan[2], Lin Wu[1], Emmanuel Arzoumanian[1], and Philippe Bousquet[1]

[1]Laboratoire des sciences du climat et de l'environnement (LSCE/IPSL), CNRS-CEA-UVSQ, Université de Paris-Saclay, Centre d'Etudes Orme des Merisiers, Gif-sur-Yvette, France
[2]CEREA, Joint Laboratory École des Ponts ParisTech / EDF R&D, Université Paris-Est, Champs-sur-Marne, France

## Abstract

This study presents a new concept for estimating the pollutant emission rates of a site and its main facilities using a series of atmospheric measurements across the pollutant plumes. This concept combines the tracer release method, local scale atmospheric transport modelling and a statistical atmospheric inversion approach. The conversion between the controlled emission and the measured atmospheric concentrations of the released tracer across the plume places valuable constraints on the atmospheric transport. This is used to optimize the configuration of the transport model parameters and the model uncertainty statistics in the inversion system. The emission rates of all sources are then inverted to optimize the match between the concentrations simulated with the transport model and the pollutants' measured atmospheric concentrations, accounting for the transport model uncertainty. In principle, by using atmospheric transport modelling, this concept does not strongly rely on the good colocation between the tracer and pollutant sources and can be used to monitor multiple sources within a single site, unlike the classical tracer release technique. The statistical inversion framework and the use of the tracer data for the configuration of the transport and inversion modelling systems should ensure that the transport modelling errors are correctly handled in the source estimation. The potential of this new concept is evaluated with a relatively simple practical implementation based on a Gaussian plume model and a series of inversions of controlled methane point sources using acetylene as a tracer gas. The experimental conditions are chosen so that they are suitable for the use of a Gaussian plume model to simulate the atmospheric transport is appropriate. In these experiments, different configurations of methane and acetylene point source locations are tested to assess the efficiency of the method in comparison with the classic tracer release technique in coping with the distances between the different methane and acetylene sources. The results from these controlled experiments demonstrate that when the targeted and tracer gases are not well collocated, this new approach provides a better estimate of the emission rates than the tracer release technique. As an example, the relative error between the estimated and actual emission rates is reduced from 32% with the tracer release technique to 16% with the combined approach in the case of a tracer located 60 metres upwind of a single methane source. Further studies and more complex implementations with more advanced transport models and more advanced optimizations of their configuration will be required to generalise the applicability of the approach and strengthen its robustness.

# 1   Introduction

Atmospheric pollution due to anthropogenic activities is a major issue both for air quality and for climate change. Industrial sites are known to emit a significant part of the pollutants and greenhouse gases. For instance in France, industrial emissions represent between 10 and 30% of major air pollutants, such as carbon and nitrous oxide (Bort and Langeron, 2016). Currently, industries must list their emissions through national inventory reports, and some of them commit to reducing these emissions. However, the choice of an appropriate mitigation policy and the verification of its results require a good understanding of the emitting processes and a precise quantification of the emission rates. Industrial emissions are difficult to model and quantify because of the diversity and the temporal variability of the emitting processes.

Many emitting industrial sites have a typical size of 100 $m^2$ − 1 $km^2$, and they emit pollutants from very specific locations within this area. Once emitted from a single or multiple point sources, the transport of these pollutants in the atmosphere over distances from 0.1 to several kilometres forms a distinct plume or multiple plumes eventually merging at larger distance downstream. One approach developed to quantify the emissions from such sites involves atmospheric concentration measurements around the site, particularly across these emission plumes, and proxies of the atmospheric transport. These proxies are used to characterise the link between the emission rate and the structure and amplitude of the emission plume. The "inversion" of this link enables the estimate of the emission rates from the observed concentrations. Among the different techniques to estimate emissions from concentrations is the tracer release method. It is often realised in the form of mobile continuous measurements across the emission plumes of the studied pollutant and of a tracer purposely emitted with a known rate as close as possible to the suspected pollutant source (Lamb et al., 1995). In this method, the proxy of the atmospheric transport is given by the relation between the tracer emission rate and the tracer concentrations. In practice, it provides estimates of the emissions of a site over a relatively short time window, i.e., typically few hours during a given day, which generally corresponds to the time during which the tracer can be released or mobile measurement can be conducted.

This approach is relatively simple to implement and enables instantaneous estimations for a large number of sites. Nevertheless, this technique encounters some limitations, particularly (i) when it is difficult to position the tracer emission close to the sources, (ii) when the sources are spread over a significant area compared with the distance between the sources and the location of the measured concentrations, or (iii) when targeting individual estimates of the different emission rates from multiple sources whose plumes overlap at a given site at the distances at which the measurements can be conducted (Mønster et al., 2014; Roscioli et al., 2015). Typically, in industrial sites, pollutant sources may be sporadic and diffusive over a large area, their location can be difficult to reach and the spatial distribution of the emissions is not always precisely known, e.g., when considering transitory leakages or widespread and heterogeneous sources. In these cases, the tracer release method can induce errors in the flux estimation since the tracer plume by itself cannot be used as an accurate proxy of the local transport from the targeted gas sources to the measurement locations. Moreover, this approach can hardly be used to provide an estimate of the different sources within a site.

Other techniques exploit atmospheric measurements using local atmospheric dispersion models to simulate the transport of the targeted gas from its sources to the measurement locations (Lushi and Stockie, 2010). Micrometeorological measurements are often conducted in parallel with those of the targeted gas concentrations in order to support the set-up of such models (Flesch et al., 2004, 2007; Gao et al., 2009). In theory, the model and the inversion of this proxy of the atmospheric transport can be applied for a point source or for a source whose spread is known. In principle, they can also be applied to multiple sources. The principle of this technique is relatively simple, but the representation of the emission spread in these models, the simulation of the transport by these models (even when they are constrained using micrometeorological measurements), and the separation of the different plumes associated with the different sources when targeting multiple sources can bear large uncertainties. In particular, the transport over short distances or time scales in a complex terrain can be characterised by complex turbulent structures which are difficult to match with a model even when the underlying processes are taken into account. Moreover, when targeting several sources, this technique relies on the mathematical inversion of a square matrix characterising the atmospheric transport that links the set of sources to the observation data. This artificially requires extending or limiting the number of observation data from the measurement series to the number of sources to be quantified. It can lead to a loss of

information or it can hide the fact that the problem is underconstrained when the plumes overlap too much.

The statistical inversion framework, which can be viewed as a generalised inversion technique, can account for uncertainties in the model. It can also address under- or overconstrained mathematical problems when constraining the source estimation with the correct number of observation data that corresponds to the complementary pieces of information in the measurements. In such a framework, a statistical estimate of the emission rates for the different targeted sources is derived to optimize the fit to the measurements, accounting for the statistical uncertainties in the source and transport modelling, in the measurements and in the prior knowledge about the source location and magnitude (Goyal et al., 2005). Statistical inversions using atmospheric transport models and atmospheric concentration measurements have been used for decades to infer surface sinks and/or sources of pollutants and greenhouse gases at the continental to the city scales (Gurney et al., 2003; Bréon et al., 2015). However, the skill of such approaches strongly relies on a good accuracy of the transport modelling and on the ability to characterise the statistics of the modelling uncertainties. It can also strongly rely on the prior knowledge emissions, in particular on the spatial distribution of the multiple sources within an industrial site for the type of applications considered in this study, and on the ability to characterise the uncertainties in such a knowledge.

This study describes a concept which combines the tracer release technique, local scale transport modelling and the statistical inversion framework to improve the estimation of gas emissions from one or several point sources in an industrial site-scale configuration. It is based on the same measurement framework as the tracer release technique. It consists of using the knowledge on the transport given by the tracer controlled emission and concentration measurements to optimize the calibration of the transport model parameters and to assess the statistics of the model errors for the configuration of the inversion system. A practical implementation with a Gaussian plume model is demonstrated and its robustness is evaluated to illustrate the principle and the potential of the concept. This practical implementation is tested for the quantification of methane emissions during a time window of several hours using acetylene as a tracer gas and mobile measurements across the methane and acetylene plumes.

Methane is an important greenhouse gas with poorly known point source emissions (Saunois et al., 2016). Typical methane emitting sites due to anthropogenic activities include waste processing plants (wastewater treatment plants and landfills), oil and gas extraction and compressing sites and farms (Czepiel et al., 1996; Yver Kwok et al., 2015; Marik and Levin, 1996). Such sites contain widespread and heterogeneous sources (like the basins in waste water treatment plants, the cells in landfills and the livestock in farms) and are prone to fugitive leakages (especially in the oil and gas sectors). Until recently, there were no strong incentives to estimate site emissions using dedicated measurements. The reported estimates were usually derived using standard bottom-up product of emission factors times quantity of waste/wastewater/gas processed and/or relatively simple emission models (IPCC, 2013). However, a precise estimate of the methane emissions from such sites based on atmospheric techniques could help their operators in their local action plans to mitigate their emissions in the context of climate change. Instantaneous estimates of the emissions through a dedicated measurement campaign can help to detect and provide a useful order of magnitude for such sources that are generally poorly known (Yver Kwok et al., 2015). The results from series of campaigns can be extrapolated to estimates for long timescales. However, a continuous monitoring of such emissions with permanent measurements would help characterise the dependence of such emissions on meteorological conditions and on changes in the site processes through time.

Here, we conduct a series of controlled experiments with known emissions of methane from one or two sources and of acetylene from one source, in meteorological and topographical conditions that are compliant with the use of a simple Gaussian plume model. Concentrations are measured through the methane and acetylene plumes at an appropriate distance from the source, as described below. The known emission of methane is used to validate the inversion results and thus to assess the efficiency of our inversion system. In particular, the fit between these results and the actual emissions is compared with the one obtained with the more traditional application of the tracer release technique to demonstrate, in our experimental conditions, the asset of the statistical inverse modelling framework. In section 2, we detail the theoretical framework of the tracer release technique, the local dispersion modelling, the statistical inversion, and our concept that combines these different techniques. We also give some practical details regarding their application to the monitoring of methane sources, and regarding the use of a Gaussian plume model for suitable meteorological and topographical conditions. Then, we describe

the configuration and the results of the experiments conducted in this study to evaluate the potential of our approach (section 3). The results and perspectives of the study are discussed in section 4 and 5.

# 2 Methods

## 2.1 Instantaneous quantification of pollutant sources using mobile measurements across the atmospheric plumes

The presentation of the atmospheric monitoring techniques below focuses on their specific configuration for the quasi instantaneous estimation of emission rates from gas sources within a targeted site. These techniques apply to gases that can be considered inert (non-reactive) on the relevant atmospheric transport and mixing timescales of the experiment. In this case, the representation of atmospheric transport, linking the emissions to the gas concentrations, can be considered linear. Given that these timescales typically correspond to 1 to 10 hours, it applies to most pollutants in practice. In this configuration, several times over the course of a few hours and at an appropriate distance from the site, the concentrations are measured along transect lines across the plumes of a gas emitted by the sources. The emission plumes are associated with an increase of the gas concentrations above the "background" concentration. This background concentration can be characterised by the gas concentrations in the vicinity of the measurement locations that has not been affected by the sources. The increase above the background concentration is proportional to the emission rates (due to the linearity of the atmospheric transport) and it can be identified in the measurements across the plume. Ideally, there should be no other major gas emitter in the vicinity of the targeted site to ensure that, due to the atmospheric diffusion over long distances, the concentrations upwind the site are relatively constant. In such conditions the background concentration can be easily characterised.

The choice of the measurement distances should follow several criteria. On one hand, the distance has to be large enough such that the transport from the source to the measurement is correctly characterised with a local scale transport model or the proxy from the tracer release. This distance depends on the spread of the single or multiple targeted sources and thus indirectly on the size of the industrial site, but also on the meteorological conditions like the wind speed and the atmospheric stability. On the other hand, it should be short enough such that the amplitude of the measured concentrations is high enough compared to the measurement and model precision. This criteria essentially depends on the emission rates due to the linearity of the atmospheric transport from the sources: the larger the source, the larger the ratio of the signal to the noise of the measurement, modelling and background and thus the higher the precision of the inversions. Finally, the choice of the distance is constrained by the need for conducting measurements on roads located downwind of the site sources (depending on the specific wind directions during the measurement campaigns) when using instruments onboard cars as in this study.

The simulated relation between the gas emission rates from the single or multiple sources of the site and the atmospheric concentrations relies on the knowledge of the location and spread of each source and on the proxy of the atmospheric transport. It is linear and expressed by the observation operator $\mathbf{H}$. The relation between the measurement indices of the concentration increase in the emission plumes or "plume indices" hereafter, called the observation vector $\mathbf{p}$, and the targeted emission rates, called the control vector $\mathbf{f}$, is given by the observation equation:

$$p = Hf + \varepsilon_0 \tag{1}$$

$\varepsilon_0$ represents the sum of errors from the observation operator, in the measurements and in the estimate of the background concentration. The observation vector is derived from the gas concentrations measured for each cross-section of the gas plume(s). The atmospheric transport proxy can be derived using the relationship between the known tracer emission collocated with the targeted sources and the tracer concentrations in the tracer release technique (section 2.2) or using a local scale atmospheric transport model (section 2.3). Inferring gas emissions from gas concentrations implies inverting the atmospheric transport to express $\mathbf{f}$ as a function of $\mathbf{p}$. If the size of $\mathbf{f}$ is the same as that of $\mathbf{p}$, i.e. if the number of plume indices derived from the concentration measurements is set equal to the number of targeted sources, the atmospheric transport matrix $\mathbf{H}$ is a square matrix. If $\mathbf{H}$ is mathematically invertible, i.e., if the problem is not under constrained due to using indices on plumes that overlap too much, and if the measurement, background and observation operator errors $\varepsilon_0$ are ignored, $\mathbf{f}$ can directly be derived

as $\mathbf{H}^{-1}\mathbf{p}$ (sections 2.2 and 2.3). If the sizes of $\mathbf{f}$ and $\mathbf{p}$ differ, or if the measurement, background and observation operator errors $\varepsilon_0$ are to be accounted for, statistical inversion approaches can be performed to retrieve an optimal estimate of $\mathbf{f}$ (sections 2.4 and 2.5).

## 2.2  The tracer release method

The tracer release method was developed to quantify pollutant emissions and has already been used in a wide range of studies to estimate the sources of various types of gases such as methane (Babilotte et al., 2010), carbon monoxide (Möllmann-Coers et al., 2002) and isoprene (Lamb et al., 1986). This method consists of releasing a tracer gas with a known rate close to the targeted gas source when this source is clearly localised and of measuring both the targeted and tracer concentrations in sections of the downwind emission plumes. When targeting the total emissions of a site with multiple sources, the tracer release is generally located in the middle of these sources, assuming that the site is seen as a point source from the measurement locations.

When both the released tracer and targeted sources are perfectly collocated and constant in time, the tracer and targeted gas concentrations have the same spatial and temporal relative variations in the atmosphere, i.e., the plumes of the targeted gas and of the tracer have the same structure. In such a configuration, the knowledge of the ratio between the tracer plume index $p_t$ and the tracer controlled emission rate $f_t$ provides a perfect (scalar) observation operator h. It thus provides a perfect estimate of the ratio between the targeted gas plume index $p_m$ and the targeted gas emission rate $f_m$. By ignoring the measurement and background errors, the targeted emission rate can be estimated using the following formula:

$$f_m = f_t \times \frac{p_m}{p_t} \tag{2}$$

Various types of plume indices p can be used (provided that they are consistently derived for the tracer and targeted gas). The background concentration is generally derived from the measurements before and after crossing the plumes. Then, the plume indices can typically be calculated using the difference between the maximum concentrations (peak heights of the signals) and the background concentration. It can also be derived from the areas between the plume signals and the background concentration (Mønster et al., 2014). When the sources of the released and targeted gases are perfectly collocated and when their emission rates are constant, both of these approaches provide the same result given that the tracer and targeted emission plumes have the same structure. However, if the collocation of both sources is not perfect or if the targeted emissions vary in time, then the shapes of the emission plumes of the released tracer and of the targeted gas can differ. To minimize the impact of this difference, the ratio of the integrated plumes is generally chosen because this index is less sensitive to the impact of thin turbulent structures than the peak height ratio (Mønster et al., 2014). Other indices have also been tested to overcome this issue like the slope of the ratio between the targeted and released concentrations above the background (Roscioli et al., 2015).

The measurement transects through the emission plumes and the computation from equation 2 are generally repeated several times, typically $n_{tr} = 10$–15 times over an hour window. The mean and standard deviation $\mathrm{STD}_{tr}$ of the $n_{tr}$ different results are used as the best estimate and uncertainty assessment for the source quantification. Of note is that strictly speaking, the exact quantification of the uncertainty in the mean estimate should be $\mathrm{STD}_{tr}/\sqrt{n_{tr}}$, which will be used here, even though $\mathrm{STD}_{tr}$ is often used (Yver Kwok et al., 2015). Such statistics allow to account, at least partly, for the potential temporal variations of the emissions, for the measurement and background errors, and for the potential impact of the non-perfect collocation of the sources in the selected measurement transects. In order to strengthen the precision of the best estimate, measurement transects with low correlations between the targeted gas and the released tracer are often ignored. The reason is that such low correlations are related to critical sources of estimation errors. It can be due, for a range of local meteorological conditions, to relatively high background and measurement errors compared to the measured signal. It can also be due to a strong difference between the structures of the tracer and targeted gas plumes arising from the fact that the tracer emission is not perfectly colocated with the targeted gas emission.

A mislocation of the tracer source far from the targeted source or its location close to a targeted source whose spread is large compared to the distance to the measurements can also generate significant biases in the series of computations. Such biases can impact the average estimate of the source without

being reflected in the standard deviation of the individual emissions computations nor in the correlation between the tracer and targeted gas concentrations. The impact of the mislocation of the tracer source can be decreased by increasing the distance between the sources and the measurements (Roscioli et al., 2015) but the choice of this distance is often constrained by other considerations as discussed in section 2.1. Approaches based on atmospheric transport models have been used to account for errors arising from this mislocation (Goetz et al., 2015).

Moreover, the tracer release technique provides an overall estimate of the emissions of a site. However when the site has several sources located quite close to each other, it can hardly be used to provide individual estimates of these sources. More specifically, even with the use of different tracer release points, the technique in itself hardly provides solutions to separate overlapping tracer or targeted gas plumes associated with different point sources.

## 2.3  Using local scale transport models

Many types of transport models are used to simulate the dispersion of pollutants at the local scale, i.e. typically over distances from a few metres to 1 or 2 kilometres, from simple Gaussian models to the coupling of Lagrangian and sophisticated CFD (Computational Fluid Dynamics) models that allow to determine turbulent patterns for complex terrain through an explicit representation of reliefs and obstacles (e.g. Baklanov and Nuterman, 2009; Hanna et al., 2011). Beyond the large range of possible model complexity, a common feature of these transport models is their ability to represent sources of any geometry. Therefore, the local scale transport models allow for addressing multiple sources or sources with a significant spread far better than proxies based on collocated tracers. In a configuration similar to that of the tracer release technique where concentration measurements are conducted across the N plumes of N targeted gas sources, the local dispersion models can be used to infer the linear relationship between the emission rates and plume indices in each of the measurement transects. The models are run with a zero background concentration unless a strong signal from neighboring sources outside the targeted site need to be accounted for, which is not the case in this study.

In practice, for a given measurement transect, simulations with such models for each individual source (ignoring the other ones), with a unitary emission rate can be used to compute each column of the $\mathbf{H}$ matrix in equation 1. If the plumes of the N sources do not overlap too much and are all discernable in the measurement transect, an appropriate selection of N plume indices can be used to disentangle these different sources. In such cases, $\mathbf{H}$ is invertible and the derivation of $\mathbf{H}^{-1}$ from matrix $\mathbf{H}$ is straightforward. Consequently, if ignoring the measurement, background and observation operator errors $\varepsilon_0$, $\mathbf{H}^{-1}$ can be directly used for the inversion of the emission of the N different sources as a function of the N plume indices for each measurement transect:

$$\boldsymbol{f} = \boldsymbol{H}^{-1}\boldsymbol{p} \tag{3}$$

As with the tracer release technique, statistics of the results from multiple inversions associated with the different measurement transects can be used to derive a best estimate and its uncertainty. The correlations between the modeled and measured concentrations along the measurement transects can be used to select the most robust inversion cases.

However, the local scale transport models can bear large uncertainties that are ignored by this inversion. These errors can be directly projected onto the estimate of the emissions through equation 3, and thus strongly weaken the confidence in the results. Furthermore, such an inversion can hardly account for the amount of useful information provided by the measurements. Typically, limiting the number of plume indices to the number of targeted sources prevents from analyzing the shape of each emission plume. Such a shape can be an indicator of the measurement, background and observation operator errors $\varepsilon_0$ which can highly impact the inversion results. Finally, with such an inversion, the level of separation between the source plumes has to be evaluated before defining the number of sources that can be inverted separately within a site without solving for an under-constrained problem. When this level of separation is weak, the inversion finds a mathematical solution to equation 3 that can be highly uncertain. The lack of flexibility of such an inversion is thus problematic.

## 2.4 Statistical inversion

The statistical inversion techniques can address the issues associated with equation 3 that are discussed above. The Bayesian principle of statistical inversion is to update prior statistical knowledge (i.e. a prior estimate $\mathbf{f}^b$ and the uncertainties in it) of the emission rates $\mathbf{f}$ with statistical information from observations $\mathbf{p}$. This update accounts for the statistical uncertainties in the observations (here the measurement and background errors) and in the observation operator $\mathbf{H}$ (Tarantola, 2005). In order to account for several sources within a site, the statistical inversion needs to rely on a local scale transport model to derive the $\mathbf{H}$ matrix. This theoretical framework allows for a control vector $\mathbf{f}$ and an observation vector $\mathbf{p}$ with different sizes to be taken into account. All sources can thus be "inverted" even if there is not enough information to separate the plumes of some of them. Furthermore, the system can make use of all the information in the measurements to filter the measurement, background and observation operator errors and any signal from the different emissions plumes associated with the different sources.

Assuming that during the measurement campaign the source emission rates are constant, this framework can also be used to assimilate the data from all plume transects to compute the optimal estimate of the emission rates at once. In such a case, the observation vector $\mathbf{p}$ gathers plume indices from all the measurement transects and the $\mathbf{H}$ observation operator represents the transport, with various meteorological conditions, from the sources to all the transects. This combination presents advantages over repeating computations for each measurement transect and deriving statistics for the emission estimates out of the ensemble of computations as for the other techniques presented above. In particular, this helps accounting for the fact that the sources of errors do not have the same statistical distribution, e.g., amplitude for each transect. The previous techniques require a selection of the cases when the confidence in the observation operator is good enough to enhance the robustness of the average. By assigning model and measurement uncertainties as a function of the measurement transect and/or meteorological conditions, the statistical inversion allows the information from each transect to be weighted differently according to its uncertainty.

The prior estimate of the emission $\mathbf{f}^b$ has to be independent of the atmospheric observations and can be provided by expert knowledge, emission inventories or process-based models. In practice, it is generally assumed that the uncertainties in $\mathbf{f}^b$, in the observations $\mathbf{p}$ and in the observation operator have unbiased and Gaussian distributions. The prior uncertainty and the sum (henceforth called observation error) of the uncertainties in the observations $\mathbf{p}$ (from the measurement and background errors) and on the observation operator $\mathbf{H}$ are thus characterised by their covariance matrices $\mathbf{B}$ and $\mathbf{R}$, respectively. Following these assumptions, the "posterior" statistical distribution of the emission rate knowing $\mathbf{f}^b$ and $\mathbf{p}$ is Gaussian and is characterised by its optimal estimate $\mathbf{f}^a$ and its covariance matrix $\mathbf{A}$ given by equations (Bocquet, 2012):

$$f^a = f^b + BH^T(R + HBH^T)^{-1}(p - Hf^b) \tag{4}$$

$$A = (B^{-1} + H^T R^{-1} H)^{-1} \tag{5}$$

The matrix $\mathbf{A}$ characterises the unbiased and Gaussian uncertainty in $\mathbf{f}^a$. If the plume from a source cannot be separated from the other ones, or if the observation errors on the plume indices related to this source are very large, the posterior uncertainty in this source will be large. The $\mathbf{A}$ matrix can thus be used to evaluate the level of constraint on the different sources or on their sum provided by the selection of plume indices, and the robustness of the corresponding emission estimates. One difficulty associated with this method is the need for providing a realistic estimate of the observation error statistics which in practice are difficult to evaluate. Another issue is that even if the system correctly accounts for the transport modelling errors when being well informed about their statistics, it will derive very uncertain emission estimates if these transport errors are large.

## 2.5 A statistical inversion based on tracer release and local scale transport modelling

Here, we propose a new concept for the estimation of the gas emission rates combining the tracer release method, local scale transport modelling, and a statistical inversion framework to overcome the issues associated with these different approaches and tools as discussed above. The basis of this new concept is

the statistical inversion framework described above assimilating the plume indices from all measurement transects altogether, where the **H** matrix is derived from local scale transport model simulations for each point or spread source of a targeted site and each measurement transect.

The main idea is to use the very accurate information on the atmospheric transport in the area of interest from the tracer release method to adjust parameters of the local scale transport models and to assess the statistics of the transport errors. The "optimized" transport model and the statistics of the transport errors are then used for the configuration of the observation operator and of the observation errors in the statistical inversion system outlined in the previous section. The optimization of the transport model parameters can rely on a range of methods, from a simple comparison between an ensemble of tracer simulations with different sets of parameters and the tracer measurements, to complex tracer data assimilation.

The statistics of the misfits between the tracer measurements and the model-based concentrations when using the optimal transport model configuration are used to set up the covariances of the observation (measurement, background and observation operator) errors **R**. This requires the conversion of the errors statistics for the tracer gas into statistics of the errors for the targeted gas. Therefore, the statistics of the variability of the measured tracer and targeted gas concentrations are used to normalize the transport errors for the two species as "relative errors", and the assumption is made that the relative transport errors are the same for both species.

This optimization of the model parameters and/or characterization of the transport errors can be performed for each individual crossing of the plume or for all plume crossings together. The use of a specific optimization of the model for each plume crossing may be preferable if the local meteorological conditions evolve rapidly. Using general statistics of the tracer model-data misfits from all plume crossings would prevent weighting the transport error and thus the information for each plume crossing depending on the modelling skills. Deriving different transport errors for each plume crossing requires the extrapolation of the single set of tracer model-data misfits into statistics for each plume crossing. These different options need to be chosen depending on the experimental case.

In order to investigate the potential of this approach in a first real test case, we propose a relatively simple practical implementation using a Gaussian transport model. CFD models remain sophisticated tools. The choice of a Gaussian plume model is more appropriate for the introduction and first test of our concept but we are aware that it restrains the range of situations that can be investigated.

## 2.6 Practical implementation for the monitoring of the methane sources using a Gaussian plume model and acetylene as tracer

### 2.6.1 The Polyphemus Gaussian plume model

Gaussian plume models provide a stationary and average view of the pollutant plumes driven by meteorological conditions that are stationary in time and homogeneous in space. This is a decent approximation for the dispersion over 1–2 minutes (i.e. the typical timescale associated with our experiments) and an area of approximately 1 km$^2$ when the wind speed is relatively high. These models cannot account for the effects of complex local topography and buildings. However, they are suitable for many configurations of industrial sites located in nearly flat suburban to rural areas, and they are easily set up and applied for the simulation of local-scale transport.

In this study, the Gaussian plume model of the Polyphemus air quality modelling system (Mallet et al. (2007) http://cerea.enpc.fr/polyphemus/) is used because it has been proven to be adapted for estimating gas emissions from local sites (Korsakissok and Mallet, 2009). Gaussian plume models are based on a simple formula that provides the concentration of the pollutant at a location generated by a point source depending on the weather conditions. The Gaussian plume formula is expressed as:

$$C(x, y, z) = \frac{Y}{2\pi\sigma_y\sigma_z\bar{u}} exp\left(-\frac{(y - y_s)^2}{2\sigma_y^2}\right)$$
$$\times \left[exp\left(-\frac{(z - z_s)^2}{2\sigma_z^2}\right) + exp\left(-\frac{(z + z_s)^2}{2\sigma_z^2}\right)\right] \tag{6}$$

where C is the concentration of the pollutant at a location of coordinates (x,y,z), Y is the source emission rate, and $\bar{u}$ is the wind speed. In this formula, the x axis corresponds to the wind direction, $y_s$ is the

pollutant source ordinate (for a single source usually set to zero) and $z_s$ is the release height above the ground. As both studied gases are poorly soluble and chemically inert for the considered dispersion time scale, it is appropriate to neglect the mass loss due to dry deposition and assume a total reflection from the ground as expressed by the last exponential term in the equation. The values $\sigma_y$ and $\sigma_z$ are the horizontal and vertical Gaussian plume standard deviations and characterise the atmospheric conditions during the measurements. The modelled concentrations are strongly dependent on these two parameters. Within the Polyphemus system, several ways to parameterize these constants are available: the Doury formulas (Doury, 1976), the Pasquill-Turner formulas (Pasquill, 1961) and the Briggs formulas (Briggs, 1973).

The parameterization according to Briggs is the most flexible one. This parameterization considers the stability of the atmosphere via the six classes of Pasquill classification from A (extremely unstable) to F (extremely stable) by taking into account wind speed and solar irradiance. It also considers the type of environment with an urban mode when the emission source is surrounded by buildings and a rural mode for isolated sites (by changing the roughness factors). The standard deviations with this parameterization are given by the following equations:

$$\sigma_y = \frac{\alpha x}{\sqrt{1 + \beta x}} \quad \text{and} \quad \sigma_z = \alpha x (1 + \beta x)^\gamma \tag{7}$$

where x is the downwind distance from the source and $\alpha$, $\beta$ and $\gamma$ are coefficients that are dependent on the stability classes. All these coefficients can be found in Arya (1999).

Different source spatial extensions can also be created in this model. However, its configuration imposes the emission $f_i$ of a given source to be spread homogeneously over its extension. The Gaussian plume model cannot represent the instantaneous turbulent structures at fine spatial and temporal scales but rather represents a time-averaged view of a plume. Therefore, it is expected that by using a high number of measurement transects, the Gaussian plume model should be appropriate for catching such an average plume and that the transient turbulent patterns in the measurements would generate a sort of noise on the emission estimates without biasing it.

### 2.6.2 Adjustment of the stability class underlying the Briggs parameters and estimate of the Gaussian model errors using the tracer data

The application of the new statistical inversion strategy described in section 2.5 in connection with the Polyphemus Gaussian transport model relies on the optimization of the stability class underlying the Briggs parameters and of the plume direction as a function of the tracer measurement transects. Since wind was measured directly in our experiments, a correction of the Gaussian plume direction should not be needed, but the section 3.2 will describe practical issues which require such a correction.

For each measurement transect, the method consists in running multiple model tracer simulations with different stability classes. They are all forced with the known tracer emission rate. The model plume direction is adjusted so that the measured and simulated plumes are aligned. The stability class whose simulation of the tracer concentrations best fits the measurements is taken as the optimal one. The fit is quantitatively checked for the plume indices chosen for the definition of $\mathbf{p}$, but it is also checked in a qualitative way by analyzing the shape of the modelled and measured signals. The estimate of the Gaussian model errors is based on statistics of the misfits between the modeled and measured tracer plumes indices.

### 2.6.3 Monitoring of the methane sources using controlled release of acetylene

This method is tested for the quantification of methane sources using acetylene as a tracer gas. The lifetimes of methane and acetylene are approximately 10 years and 2–4 weeks, respectively (Logan et al., 1981). Both of these gases can be considered inert at the time scale corresponding to the time between the release of molecules at the source and the measurement of concentrations downwind in the plume.

In this study, the methane and acetylene concentrations are measured in a continuous manner along a line crossing the emission plumes using an sensitive analyser placed in a car. Our preliminary analysis shows that we obtain satisfying results when concentrations are typically measured at a distance of 100 to 1000 metres from methane sources of 1500 to 100000 $gCH_4.h^{-1}$ and spread within an area of $100 \times 100$ $m^2$ to $500 \times 500$ $m^2$.

# 3    Evaluation of the concept with controlled release experiments

## 3.1    General principle of the controlled experiments

The following sections describe the experiments under controlled conditions for both acetylene and methane used to evaluate the statistical inversion framework detailed in section 2.6 and more generally to give insights in the potential of the approach proposed in this study in section 2.5. A campaign was organized during two days of March 2016 at the Laboratoire des Sciences du Climat et de l'Environnement (LSCE) in France (longitude: 48.708831°, latitude: 2.147613°, altitude asl: 163 m). The experimental conditions were selected to be favorable for the use of a Gaussian plume model to simulate the atmospheric transport. One or two methane sources and one acetylene point release were generated with cylinders in the parking lot of the LSCE, which is located in a rural area in the southern region of Paris. The topography of this area is very flat, and only few low buildings can potentially influence the atmospheric transport from the parking lot to the road where the concentrations are measured. This road is located approximately 150 metres away from the controlled sources. No major methane or acetylene sources in the vicinity of the LSCE could disturb the measurements. Each measurement day was selected by taking the weather forecast into account and choosing days with a strong enough wind from the north to be able to measure the emissions from the parking lot on the measurement road. The average weather conditions of each measurement series are summarized in table 1.

During this campaign, the methane and acetylene sources were dispersed in four different configurations to estimate the accuracy of the proposed method and the uncertainties depending on whether the tracer gas is perfectly collocated with the methane source or not. For each configuration, the methane and acetylene emission plumes were crossed 20–40 times (see table 1), and each series of crossings was performed on the same day on a timescale of 1-2 hours. The observed increases in the acetylene and methane concentrations within the plumes ranged between 3–15 ppb and 50–500 ppb, respectively. Configurations 1, 2 and 4 were tested in the afternoons between 13:30 to 16:30 UTC while configuration 3 was tested in the morning between 10:00 and 12:00 UTC (see Figure 1).

The following sections describe the different components of the experimental and modelling systems used for the inversion of the methane sources and the results from both the tracer release technique and the combined statistical approach. These results are compared with the known methane emission rate to test the ability of each method to estimate the emissions. Statistics of uncertainties are also derived for the two methods based on the statistical frameworks described in section 2 but also based on Observing System Simulation Experiments (OSSEs) with pseudo-data.

Of note is that for the sake of clarity and simplicity, we avoid analyzing the results that would be obtained with a modelling framework where the observation operator is truly inverted (see section 2.3). Such results would not have supported the analysis of the potential of the combined approach compared to the traditional tracer release technique. Furthermore, discussing whether, in this approach, the Gaussian modelling configuration should be influenced or not by the parameter optimization based on the tracer data would be uselessly complicated.

## 3.2    Analytical equipment

Downwind gas concentrations were measured using a G2203 cavity ring-down spectrometer (Picarro, Inc., Santa Clara CA), which continuously measures acetylene ($C_2H_2$), methane ($CH_4$) and water vapor ($H_2O$). Based on infrared spectroscopy, the high precision of the system (precisions of 3 ppb and < 600 ppt for methane and acetylene, respectively, on 2 second interval) is due to its very long path length ($\simeq$ 20 km) and the small size of its measurement cell (< 35 mL). Mobile measurements with such an instrument have already been successfully performed and published in previous studies (Mønster et al., 2014; Yver Kwok et al., 2015), demonstrating the potential of this method. The measurement error encompasses the precision stated above but also the fact that the acetylene and methane are not measured at the exact same time and frequency. Indeed, acetylene is measured every second while methane is measured every other second. At the scale of our measurement (less than a minute to cross a plume), this can impact the error significantly.

Before the experiment, the instrument has been tested in the laboratory. It showed a good linearity over a large range of mixing ratios and a good stability over time with small dependency to pressure and temperature. To control for a drift, we measured a gas with a known mixing ratio (calibrated with a

multi-point calibration in the laboratory) before each series of measurements in order to ensure the good analytical performance of our instrument. Moreover, in the tracer released method and the combined approach presented in this study, we are interested in the increase of concentrations due to the tracer and targeted point sources above the background signal (i.e. the plume indices) more than in the absolute value of the measurements. Thus, an offset of the measured concentrations will not impact our estimates.

During the field campaign, wind speed and direction were taken from the meteorological station installed on the roof of the nearby laboratory at about 7 m high. The mobile system was set up in a car and powered by the car's battery. The air sampler was placed on the roof at approximately 2 metres above the ground with a GPS (Hemisphere A21 Antenna) to provide the location of the measurements. The sampled air was sent into the instrument by an external pump system allowing an inlet lag between the sample inlet and the measurements of less than 30 seconds. This more or less constant inlet lag introduced a spatial offset when comparing the measured and modelled tracer or methane concentrations. This spatial offset is the same for methane and acetylene and is well characterised by the comparison between the modelled and measured acetylene plumes. In our combined statistical approach, it is thus well accounted for when comparing the modelled and simulated methane plumes thanks to the correction of the Gaussian plume direction according to the acetylene data. Therefore, this offset is ignored hereafter.

## 3.3   Tracer and target gas release

Acetylene is commonly used as a tracer. Due to its low concentration in the atmosphere ($\simeq$ 0.1–0.3 ppb), any release is easily detected. Acetylene also presents the benefit of being relatively inert, and thus, negligible loss during the transport process is expected (Whitby and Altwicker, 1977). Other gases are suitable as tracers, such as $SF_6$, but acetylene is preferred because it is not a greenhouse gas. However, due to its flammability, its use requires specific precautions.

An acetylene cylinder (20 L) containing acetylene with a purity > 99.6% was used as the tracer source. A methane cylinder (50 L) with a purity of 99.5% was used for the controlled methane release. The flows of both gases were controlled by a 150 mm flowmeter (Sho-rate, Brooks) able to measure fluxes between 0 and 1500 L.min$^{-1}$. The different acetylene and methane emission rates were checked by weighing the cylinders before and after each test and timing the release duration. The flow rate calculated with the mass difference was systematically in good agreement with the flow rate read on the flowmeter since their relative difference was between 1 and 3 %. Therefore we believe that there was no important variability of the acetylene and methane release during our experiments. The amount of acetylene emitted was adjusted such that its emission plume can be detected on the roads where the measurements were performed while keeping it at the lowest rate possible to limit the risks associated with its flammability. In this study, we used emission rates from 65 to 90 g.h$^{-1}$ for acetylene. During the measurement campaign, the cylinders were attached with straps to a fixed frame to avoid any accidents.

## 3.4   Tested configurations of the gas releases

This section details the four configurations used during this campaign (figure 2). The first configuration consisted of a collocated emission of acetylene and methane. This configuration enabled us to estimate the accuracy of the method and our system under optimal conditions. One cylinder of methane and one cylinder of acetylene were placed on the parking lot and connected together by a tube with a length of a few metres. This system aimed at ensuring optimal mixing of both gases and was designed to be as close as possible to the ideal situation in which methane and acetylene are emitted at the same location and under the same conditions. In principle, under such conditions, the tracer concentration to emission ratio should provide a perfect proxy of the methane transport and the tracer release technique should provide better estimates than the statistical inversion that relies on an imperfect, although optimized, modelling of the methane plume. Still, both techniques should be hampered by measurement and background errors.

In reality, in industrial sites, methane source locations are not always well known, or it may be difficult to access these sources and place a tracer cylinder next to them. The second and third configurations tested the impact of non-collocated emissions of tracer and methane. To represent this situation, one cylinder of methane and one cylinder of acetylene were used, and the methane cylinder was moved i) approximately 60 metres downwind from the acetylene bottle location (second configuration) and ii) approximately 35 metres laterally compared with the wind direction (third configuration). During these

two experiments, the wind was blowing from the North, i.e. it was perpendicular to the measurement transects along the road, south of the sources.

Finally, within real industrial sites, several sources of methane may be encountered. The fourth configuration tested the influence of having two methane sources on the estimation of their fluxes when one tracer source is used. With this configuration, we also evaluate the ability of the combined statistical approach to estimate the emissions for each individual methane source. For this purpose, a system of two tubes was connected to the methane cylinder, splitting its exhaust into two locations approximately 35 metres apart. During this experiment, the wind was blowing from the North-East, i.e. it was not perpendicular to the measurement transects along the road. The acetylene cylinder was collocated with one of the exhausts.

The advantage of the combined method proposed in this study over the traditional tracer release technique (which relies on the collocation of the target and the tracer gas sources) to infer the total emissions from a site should be revealed in the second and fourth experimental configurations. In homogeneous meteorological conditions, and when the wind direction is perpendicular to the measurement transects, a shift of the methane sources in a direction perpendicular to the wind and parallel to the measurement transects should only result in a shift of the emission plumes along the measurement transects. It should not impact the plume indices from the measurement transects and thus the results from the tracer release technique. Therefore, in idealistic conditions, in the third experimental configuration, the tracer release technique should still provide better estimates than the combined approach. However, in practice, during experiments with the third emission configuration, neither the shift between the cylinders nor the measurement transects (along the slightly curvilinear road) were perfectly perpendicular to the wind direction, and they were not perfectly parallel between them. Therefore, the combined approach has potential to yield better estimates also in this configuration. Finally, it can provides estimate for both sources in the fourth configuration, while this cannot be achieved with the tracer release technique in our experimental framework due to the strong overlapping of the plumes from the individual sources (see section 3.8).

The time series of acetylene and methane measurements for each tracer release experiment are shown in figure 1.

## 3.5  Definition of the background concentration and of the plume indices

In this study, two different definitions of the plume indices to build the observation vector $\mathbf{p}$ are used but they are both based on the integral of areas between the concentrations within the plumes and the background concentration.

The portions of plume concentrations and of background concentrations in the measurement transects are defined "by eye". The portions of background concentration are restricted to $\simeq 5$ s before and after the plumes. In many cases, the increase of the concentrations due to the plumes is clear and the portions of plume and background concentration easy to define. However, in other cases the background variations near the plumes and the turbulent patterns at the edge of the plumes can have comparable amplitude so that the definition of these portions is more difficult (Figure 1). For each plume, the background concentration value used to compute the plume index is taken as the average concentration over the background portions of the transect.

When we investigate the tracer data or when we estimate the emission rate of a single source of methane, i.e. in configurations 1, 2 and 3, and in configuration 4 for the tracer release technique only, the plume indices are defined as the integral over the entire plume of the concentrations above background. In this case, the observation scalar p (when applying the tracer release technique to each transect) or vector $\mathbf{p}$ when conducting the combined statistical inversion by gathering data from all transect into a single vector, are denoted $p^{ent}$ and $\mathbf{p}^{ent}$ respectively.

When we estimate the emission rates of the two sources of methane with the combined approach in configuration 4, the portion of observed methane and acetylene increase within the plumes is divided into five slices of equal time (and identical for methane and acetylene). For each slice of a given transect, an index $p_{slc}$ is defined as the integral of the concentrations above the background in this slice. The observation vector $\mathbf{p}^{slc}$ gathers all these indices.

## 3.6 Optimization of the Gaussian plume model parameters

In the Polyphemus Gaussian plume model, the definition of the plume indices is consistent with the one in the measurements, and in particular it follows the same definition of the plume portions or slices along the measurement transects.

For each measurement transect, the optimization of the stability class underlying the Briggs parameterisation of this model is based on the fit to the acetylene plume index only. Since no measurements of solar irradiance were available, comparing the selected stability class to the theoretical one is not possible. According to the table of Pasquill which is used for the Briggs parameterisation, there are three stability classes that correspond to the 2 to 4 $\mathrm{m \cdot s^{-1}}$ measured wind speed during our experiments: the classes A and B and C. However, for a given wind speed, there is only two choices, A and B, or B and C. We have verified that the selected stability class is systematically consistent with these two theoretical options. Choosing one over the other can modify the simulated plume indices by a factor 2 to 3.

We also checked for each measurement transect that the model error is not too large. In some cases, the model cannot "reasonably" reproduce the observations due to the presence of large turbulent structures or due to transport conditions that are extremely unfavorable for the model (due to swift wind change or low wind conditions). In such situations, there is no Briggs stability class that allows for the model to fit approximately the acetylene plume index. Finally, we decided to remove transects from the analysis when the relative error between the modelled and measured acetylene plume indices was higher than 70%. This value of 70% is an empirical choice corresponding to very large modelling errors. All cases kept for the analysis had relative uncertainties well below this 70% threshold. In theory, the strategy of computing the statistics of the model error as a function of such misfits should ensure that the weight given to these transects in the inversion is low. However, in practice, we conservatively prefer to remove transects for which the confidence in the model is extremely low. This evaluation leads us to ignore 30% of measurement transects when applying the combined statistical approach.

Figure 3 illustrates the results of the model parameterization selection. In this example, which corresponds to the $5^{th}$ transect of the measurements for the configuration 2 when the wind speed was 2.9 $\mathrm{m.s^{-1}}$, the tracer concentrations modelled with the stability class B best fit the measured concentrations, which are represented in black.

## 3.7 Estimation of the biases of the tracer release method due to the mislocation of the tracer with theoretical model experiments

When using the tracer release technique, defining the optimal estimate of the emissions and the uncertainty in the estimate from the $n_{tr}$ selected transects respectively as the average estimate from the application of equation 2 and using $\mathrm{STD}_{tr}/\sqrt{n_{tr}}$, fully ignores any potential bias in the method. However, in our experiments, the mislocation of the tracer emission does not only generate random errors that are caught by the variations of the results between the different measurements transects. It also has a strong potential to generate a bias in the computations since the measurements are taken in a relatively narrow range of positions south of the sources. Such a problem applies to many of the tracer release experiments where the measurements are always taken from the same area (e.g., due to the need for using roads).

Here, we use OSSEs (Rayner et al., 1996; Chevallier et al., 2007) with the Gaussian plume model whose stability class is optimized with the tracer data to estimate the bias that can arise from the spatial offsets between the tracer and methane sources. The bias estimates will be used to complement the assessment of uncertainty in the results from the tracer technique, except for the first configuration of the experiments, for which there is no offset between the methane and acetylene sources. As discussed in section 3.4, the "lateral" (i.e. orthogonal to the wind direction and parallel to the road) offsets between the methane and tracer sources in the third experimental configurations is expected to have a relatively weak impact on the tracer release computations. There should be a far larger bias associated with the downwind shift of the unique methane source in the second configuration and with the complex shift of one of the methane source when the wind was not blowing perpendicular to the measurement transects in the fourth experimental configuration.

In the OSSEs, we assume that the true methane and acetylene emission rates are those used for the experiments with real data. The synthetic methane and acetylene concentrations are simulated with the Gaussian plume model forced with these emission rates and similar weather conditions as during

the campaign. The corresponding emission plume transects for both gases are extracted along the same paths as during the campaign. Finally, equation 2 is applied with the acetylene and methane plume indices from these simulations and the acetylene emission rate, and the resulting methane emission rate is compared with the actual one. The comparison provides a direct estimate of the bias associated with a spatial offset between the acetylene and methane sources since in these computations (i) stationary conditions are implicitly assumed, (ii) the same model configuration is used to simulate the acetylene and methane concentrations, and (iii) we ignore the background and measurement errors.

In the following, we characterise the biases by their absolute value and the fraction of the actual source that they represent. The bias is estimated to be 69% for configuration 2, 12% for configuration 3 and 56% for configuration 4. Considering the amplitude of these errors, we can expect that our combined statistical approach has a high potential for providing better estimates than the tracer release approach for configurations 2, 3 and 4.

Additional OSSEs are conducted to better characterise the biases as a function of the upwind or downwind shifts of the tracer source compared to the targeted source and as a function of the distance between the sources and the measurement locations. They correspond to the theoretical experimental configurations with one methane and one acetylene source only, and they use northern wind conditions as was measured during the first experimental configuration. Upwind and downwind offsets from 20 to 200 metres between the methane and tracer sources are tested with OSSEs, with hypothetical measurement transects perfectly orthogonal to the plumes (wind) directions at different distances from 100 to 2750 metres from the methane source. The corresponding estimates of biases are presented in figure 4, with the results for the downwind and upwind shifts of the acetylene source provided in figures 3a and 3b respectively.

When the tracer is released upwind of the methane source, the emission rate is overestimated because of the vertical atmospheric diffusion, which makes the integral of the released tracer concentrations through the emission plume near the ground lower than if both sources were collocated. The opposite occurs if the released tracer is placed downwind of the methane emission location. When the tracer source is either upwind or downwind of the methane source by more than 100 metres and the measurements are taken at less than 300 metres, the bias exceeds 40% . The biases due to upwind shifts are generally similar to the biases due to downwind shifts over the same distances. When the measurement distance increases, the impact of the shift between the sources decreases. When the measurements are taken at more than 1200 metres, the bias becomes less than 20%. However, at such distances, with the emission rates used in our experiments, the signal to measurement and background noise ratio would likely be too small for our instruments to derive precise estimates of the emissions.

## 3.8 Tracer release method estimates

Figure 5 presents one example of the measured acetylene and methane cross-sections used for calculating the methane emission rate for each campaign. For the first series, both the acetylene and methane profiles are similar due to the collocation and the mixing of the sources, but we can still observe a significant difference between both emission plumes due to measurement and background errors. The shift between the sources is reflected by a smaller relative amplitude and a higher relative width of the acetylene plume compared to the methane plume in configuration 2 than in configuration 1 and by a lateral shift of the acetylene plume compared to the methane plume in configuration 3. The two overlapping methane emission plumes, one superimposed with the acetylene plume, can be distinguished in the fourth configuration.

In this section, the uncertainties in the optimal (i.e., average) estimates of the sources are characterised by the random uncertainty which is given by (i) the variations of the results between the measurement transects, $\text{STD}_{tr}/\sqrt{n_{tr}}$, (ii) the bias due to the mislocation of the tracer (see section 3.7 above), and (iii) the standard deviation of the total uncertainty being taken as the root sum square of these two terms. Table 2 lists the estimated methane emission rates and the methane emission rates actually released for each tested configuration.

These results confirm that the estimates closest to the actual methane rates are obtained for the first and the third configurations with a relative difference of 14% and 11% respectively. However, surprisingly they are slightly higher for the first configuration than for the third one. Furthermore, these errors are relatively high for the tracer release technique. They are mainly due to the variations in the background concentrations for methane, but also in some cases for acetylene. For example, the methane

background can range between 2079 and 2099 ppb from one crossing to another for the first configuration or between 2012 and 2031 ppb between transects for the third configuration. Moreover, the standard deviations within the background portions used to compute the background concentration can reach 9 ppb for methane and 1 ppb for acetylene. These variations characterise the level of uncertainty in the background concentration and they are significant compared to the amplitude of the plumes. The measurement errors associated with the lagtime between the methane and tracer concentrations may also play a significant role in the level of error associated with the estimates from the tracer release technique. Instrument precision, on the other hand, should not significantly contribute to the error since its amplitude is much smaller than the typical signals measured throughout the experiments (figure 1 and 5).

The relative differences between the actual rates and the tracer release estimates are much more important for the second and the fourth configuration, 32% and 67% respectively. The comparison between these results and those estimated in the first and third configuration indicates that in the latter cases, the observation operator errors associated with the mislocation of the tracer are much more important than the impact of the measurement and background errors. These error estimations based on direct comparison of the known emission rates are relatively well reflected by the uncertainty estimates, which are much lower for configurations 1 and 3 than for the second and fourth one, both in terms of random error and in terms of biases.

## 3.9 Combined approach

### 3.9.1 Configuration of the statistical inversion parameters

In this section, we provide details on our definition of the prior estimate of the sources $\mathbf{f}^b$, of the covariance matrix of its uncertainties $\mathbf{B}$, and of the covariance matrix of the observation errors $\mathbf{R}$ that are needed for the application of equation 4 underlying the statistical inversion.

Here, we assume that the measurement and background errors are negligible compared to the transport errors, and thus that the observation errors can be represented solely by the transport errors. This assumption arose from the relatively high values taken by the transport error estimates. The modelled vs. measured tracer plumes indices, and their product with the ratio between the measured methane and tracer plume indices are thus used to set up the variances of the observation error in the inversion configuration, i.e., the diagonal of the covariance matrix $\mathbf{R}$. In the case of a unique methane source, we use the absolute value of the difference between the modelled vs. measured plume indices to define the standard deviation of the observation error for the corresponding transect. When there are several methane sources within a site, we use the absolute value of the difference between the modelled vs. measured plume indices for each slice of the measurement transects (see section 3.5). We assign a minimum value for these standard deviations to prevent one transect or slice of a transect to dominate too much over the others in the inversion process. In the least squares minimization process associated with the statistical inversion, a data assimilated with a considerably lower observation error than the others may fully drive the inversion results. For some transects, an excellent fit may occur between the model and the measurements in terms of plume indices (i.e., integration of the emission plume concentrations over the background) despite the shapes of the modelled and measured tracer plumes being significantly different, revealing some significant observation errors. Applying a threshold to the observation errors limits the impact of their underestimation through the objective comparison between the modelled and measured plume indices. We make the assumption that there is no correlation of the transport errors, and thus of the observartion error (assuming that it is dominated by the transport errors) from one slice to the other slice of a given transect or from one transect to another one such that the $\mathbf{R}$ matrix is set up diagonally.

The typical prior knowledge $\mathbf{f}^b$ on the emission rate, from waste treatment sites, farms, or gas extraction or compression sites from process models, typical national- to regional-scale factors is generally highly uncertain. It can bear more than 100% uncertainty and for many of these sites, the order of magnitude of the uncertainty in the emissions is not known. Despite working in the framework of a controlled release experiment, we attempt to set up the inversion system to have the same conditions as when monitoring the emissions from such sites. We thus set up the prior estimate of the methane emission rates to 1800 $g.h^{-1}$ and the standard deviation of the prior uncertainty in these rates to 80% of this prior value. This ensures that the prior has a weak impact on the results. In general, there is

no correlation between the prior uncertainties in the methane emissions from different targeted sources within a site since they generally correspond to different processes (e.g. the aeration process and the clarification process in wastewater treatment plants (Yver Kwok et al., 2015)). Therefore, here, the **B** matrix is set up diagonally.

### 3.9.2 Results

Figure 6 presents examples of results obtained using the combined statistical approach with one or several methane sources. The behaviour of the inversion system and the values in the concentration and observation space are illustrated for one transect only (for the $3^{rd}$ transect of the first configuration and for the $38^{th}$ transect of the fourth configuration). It shows that the posterior estimates of the emissions have a much better fit of the simulated concentrations and plume indices than the prior emissions.

Table 2 presents the methane emission rates estimated with the combined approach for each tested configuration. We also analyse the covariance matrix **A** of the theoretical uncertainty in the emission estimates when using the statistical approach (equation 5), which provides a complementary assessment of the reliability of the results and of the level of separability between the two methane sources when using several of them in the experiments.

For the first and the third configurations, the statistical inversion gives similarly good estimates of the methane emission rates as the tracer release method, with relative deviations from the actual rates of 15% and 7% respectively. As expected, the tracer release technique provides better results for the first configuration. However the corresponding difference or relative error is very small and the combined statistical approach provides better results in the third configuration. Furthermore, the combined approach derives relatively good estimates for the second and the fourth configurations as well, contrary to the tracer release method. Indeed, for both of these experiments, the relative differences between the actual rates and the combined approach estimates are 16% and 4% respectively. Since being impacted by the background and measurement errors, this approach sill provides relative errors around 15% for configurations 1 and 2 but they get lower than 10% for the third and fourth configuration.

In all cases, the statistical inversion predicts a very low posterior uncertainty in the emission estimates for each configuration. For the fourth configuration with two methane sources, the approach fails at deriving precise estimates of each source due to the important overlapping of their emission plumes during most of the crossings. Indeed the system attributes almost all the emissions to one of the two sources and none to the other one. The diagnostic (through the computation of **A**) of negative correlation (-0.41) of the posterior uncertainties in these two sources supports the assumption that there is a weak ability to separate the signal from each source due to their overlapping, and that it is the main source of error in their individual estimates.

## 4    Discussions

The general results from these experiments indicate that both the tracer release technique and the combined statistical inversion system can provide good orders of magnitude of the total methane emission rates for each of the four source configurations that we have considered. However, the results when using the most favorable configurations of controlled emission where the methane source is collocated (configuration 1) or nearly aligned with the tracer source in the direction orthogonal to the wind direction (configurations 3) can still bear more than 10% relative errors. This is relatively high for the tracer release technique compared to what has been obtained, e.g., by Allen et al. (2013). For both the tracer release technique and the combined statistical inversion, the best results are not obtained for the most favorable controlled emission configuration when the acetylene and methane sources are collocated. This is surprising, since in such a configuration, the acetylene should provide a very precise (perfect if ignoring the measurement and background uncertainties) proxy of the atmospheric transport of methane. In the configurations with non-collocated sources, the results in the other configurations should be hampered by larger uncertainties in the representation of the atmospheric transport due to the local variations of the wind between the methane and the acetylene sources.

Actually, the variations of the atmospheric conditions from one experimental configuration to the next reveal to be the strongest driver of the precision of the results in our study. It changes the turbulent patterns and thus the transport errors when using the model or when using the tracer with a mislocated

source. It also changes the typical amplitude of the tracer and methane signals, and thus the signal to measurement and background noise ratio. The signal to measurement and background noise ratio is critical and strongly influences the inversion precisions since for many measurement transects, our measurement and background errors appear to be significant compared to the amplitude of the measurements. The measurement precisions are a negligible source of error given the typical concentrations measured in this study. However, the small timelags between the acetylene and methane measurements are presumed to raise significant uncertainties in the comparison between acetylene and methane data. The variations in the background concentrations for methane, but also in some cases for acetylene, also prove to be high enough to raise uncertainties in the single "background value" used for the computation of the so-called plume indices, i.e. the integral of the increase of the concentrations above the background within the plumes.

We anticipate that the results would have been better and more robust if the methane emission rates had been larger due to the increase of the signal to measurement and background noise ratio. In real application cases, the methane industrial emissions are definitely higher than the controlled emissions used in our experiments and we can thus expect the issue of the measurement and background errors to be less critical. Furthermore, we ignored these errors when deriving the covariance of the observation errors in the statistical inversions while several indicators could have been used to characterise their statistics. We could thus help the combined statistical inversion system better account for them when they are significant.

Despite these issues, this set of experiments clearly confirmed our expectations regarding the tracer release technique and the combined statistical inversion. In the configuration with the methane and acetylene sources collocated, the tracer release method provides better results than the statistical inversion since the latter is impacted by significant transport errors in addition to background and measurement errors while the tracer release technique is impacted by the last two sources of errors only. The optimization of the Gaussian plume model using the acetylene data still proves to be efficient to limit the transport errors so that the accuracy of the statistical inversion is still close to that of the tracer release technique for the first experimental configuration.

In the other experimental configurations, which are representative of frequent situations in industrial sites when the tracer cannot be released close to the single or multiple targeted sources, the combined statistical inversion provides better results than the tracer release technique. Our OSSE demonstrates that the mislocation of the released tracer can induce large errors when considering moderate distances between the tracer and the targeted sources even with much larger distances between the measurements and the sources. In these cases, our experiments with real data illustrated that the calibration of a Gaussian plume model using the tracer release method and the integration of the calibrated model in a statistical inversion framework help to reduce this error. The better behavior of the statistical inversion compared to the tracer release technique cannot be explained by a stricter selection of the measurement transects by the former. We recomputed the results from the tracer release technique when limiting the selection of the transects to that of the combined statistical approach and found very similar results (33% of error instead of 32% for the second configuration). On the opposite, the need for using a stricter selection of measurement transects that fit with the Gaussian plume model can be seen as a weakness of the combined inversion approach. The reduction of the transport error when using the model rather than the tracer with a mislocated source is the best explanation for the improvement of the results with the statistical inversion. These critical results demonstrate, in practice, the potential of our new method to provide better estimates than the traditional tracer release technique.

However, our results from the experiment with the fourth configuration of the controlled emissions fails to demonstrate the skills of the atmospheric inversion for providing precise estimates of the different emission rates from the multiple sources within our site. At least, it shows that the statistical inversion could diagnose by itself, with the estimate of the posterior uncertainty covariance matrix, indications that the two targeted sources of methane were too close such that their plumes were hardly separated by the inversion in this fourth configuration.

The much lower uncertainties associated with the statistical inversion results seem to confirm that they are more robust than those from the tracer release method. However, even though the uncertainty estimates in both methods are supposed to cover all sources of uncertainties, they rely on very different assumptions regarding these sources of uncertainties and on very different theoretical derivations. In particular the statistical inversion ignores biases while we explicitly accounted for biases in the tracer release technique. Furthermore, unlike the estimate of uncertainties for the tracer release technique, the

statistical inversion ignores the variations of methane model data misfits from one transect to the other one, while these misfits could be an indicator of the uncertainties in the emissions. It strongly relies on our characterization of the transport errors and prior uncertainties. We tried to rely on an objective quantification of the transport errors and we used such a high uncertainty in the prior flux estimates that this estimate did not have a large weight in the statistical inversion. However, the derivation of the transport error still relied on strong assumptions regarding its structure, and in particular regarding its spatio-temporal correlations. Actually, the computation of transport uncertainties using model-data misfits for tracer plume indices that are integrated over the whole plumes, i.e. for the same tracer plume indices as that used to optimize the transport model configuration, raises theoretical issues in the first three experimental configurations. It assumes that the dominant source of transport model errors is related to the inability of the transport model, in the range of parameterization that are tested, to fit perfectly these plume indices. This does not account for the inability of the transport model to catch the turbulent patterns and the variations of the wind conditions in space and time. Assessing the transport model errors based on statistics of model-data misfits for slices of the tracer plume as in configuration 4 may help better account for such sources of transport errors. However, in general, the information on the transport model errors from the tracer data may have to be complemented by other sources of information on such transport errors, in addition to information about the background and measurement uncertainties (as discussed above) to avoid under-estimating the overall observation errors, and thus, consequently, the posterior uncertainties. All of this makes the comparison of the error bars for the two methods difficult and weakens the reliability on the quantification of the uncertainties in the results from the statistical inversions, especially since they appear to be very low for all experiments. These uncertainties should be used cautiously as an indicator of the relative behavior of the system rather than an absolute indicator of the result precision.

The promising results from this study should not be generalised into a comprehensive evaluation of the robustness of the concept. Here, the practical use of a Gaussian plume model is made suitable by the choice of the experimental conditions over a flat terrain and for relatively stationary and homogeneous wind conditions. Such conditions may be difficult to gather when conducting real measurement campaigns for industrial sites. This new method cannot be generalised if relying on a Gaussian plume model, while the tracer release technique is adapted to a far larger range of meteorological and topographical conditions. There is a need of studies to investigate the use of more complex types of local transport models (e.g. Lagrangian models driven by diagnostic wind flow model or 3D flow fields from CFD simulation) to apply the combined statistical approach to such a range of conditions.

Furthermore, as indicated above, the turbulent patterns induced significant transport errors that contributed to the significant uncertainties in the inversion results. The strict selection of the measurement transects that can be exploited by the inversion system is strongly related to the poor ability of the Gaussian plume model to simulate many of them. This is demanding in terms of measurements, many transects being needed to ensure that a significant set will be used for the statistical inversion.

At last, for the optimization of the Gaussian plume model settings, the variable selection of stability classes representative of less than 15 minutes measurement transects is questioning. In the method, the fit of the model to the tracer data is the only critical criteria while the consistency between the stability class and the meteorological conditions according to the Pasquill table is just checked as a diagnostic that does not have any weight in the model optimization. However, the changes in the resulting stability classes over short timescales question whether the Gaussian plume model is appropriate for such a combined inversion technique. A direct optimization of the typical diffusion length of the Gaussian plume or of the parameters of the Briggs formulation, instead of the selection of the optimal stability class underlying such parameters, would allow a better, if not perfect, fit to the tracer plume indices. However, such an optimization could increase the lack of physical consistency between the resulting model parameters and the actual meteorological conditions due to the limitation of the Gaussian representation of the plumes.

All these problems contributed to the significant errors in the statistical inversions in this study and could make such errors too large in complex cases of actual industrial emission quantification. Therefore, while the choice of the Gaussian plume model for the initial tests to evaluate our new concept was appropriate, future studies should investigate how more complex models could be integrated in this inverse modelling framework. However, the control of CFD driven dispersion models to fit the tracer data will not be straightforward even if attempting at extracting far more information from these data than simple plume indices. Even if modelling turbulent structures, the CFD models would be hardly controlled to fit that of the measurements. In general, the appropriate control techniques could be as

complex as tracer data assimilation in these models which would make the method far more difficult to apply than in our study. This increase of complexity may make the method quite difficult to apply while there is a need for precise and easy-to-implement method for estimating methane emissions from the industrial sector. From this point of view, the tracer release technique definitely appears to be the most efficient technique.

Our concept faces another type of challenge. During measurement campaigns on actual industrial sites, the locations of the methane sources are not exactly known as in our tests. This lack of information could induce additional uncertainties to our estimates. Another source of uncertainty is the fact that in the tested configurations, methane point sources were used whereas during field campaigns, spread and fugitive sources may be encountered while their spatial distribution could be poorly known. The lack of knowledge on the emission spatial distribution may decrease the advantage of the combined approach (which, in its present form strongly relies on this knowledge) compared to the release technique.

# 5    Conclusion

We propose a new atmospheric concentration measurement-based concept for instantaneous estimates of gas emissions from point sources or more generally from industrial sites. This concept combines the tracer release technique, local scale transport modelling and a statistical inversion framework. The idea is to optimize the model parameters based on the knowledge provided by the tracer release and concentration measurement and to exploit tracer model-measurement misfits to prescribe the statistics of the modelling error in the statistical inversion framework. Compared to the traditional tracer release technique, the method has the advantage of exploiting the knowledge on the atmospheric transport provided by the known tracer release and measured concentration without relying on the collocation of the tracer emission and of the targeted gas emission. This is a critical advantage since the tracer can hardly be collocated with the targeted sources in much of the real industrial cases. The statistical framework can account for the different sources of uncertainties in the source estimate, can solve different targeted sources together and can consider any valuable number of pieces of information in the measurement of the targeted gas for such an inversion.

We also propose a relatively simple implementation of this concept using a Gaussian plume model. Finally, we apply it to a series of controlled release experiments with methane and acetylene taken respectively as targeted and tracer gas and we compare its results to that of the tracer release technique to demonstrate the added benefit of our new approach. The results indicate that when the tracer and targeted gas sources are collocated, the combined statistical approach yields results that are nearly as good as that from the tracer release technique, even though, the former can be impacted by the transport modeling errors which do not apply to the latter. More importantly, the results confirm that when the tracer and targeted gas sources are not collocated, the combined statistical approach provides better results than the tracer release technique. Such results with a rather simple implementation of the combined statistical approach using a Gaussian plume model are highly promising for our concept.

Our experiments fail to demonstrate the potential of this approach to estimate the different emission rates from the multiple sources within a site. Furthermore, as highlighted by section 4, the robustness of the method and its assessment of the uncertainties need to be improved. The generalization of the method for applications to complex sites and meteorological conditions, based on more realistic transport models, will confront difficult technical and scientific challenges. However, at least, our experiments promotes further studies and development of our combined approach, and even applications of our simple implementation framework to the instant quantification of real industrial sites emissions for which the conditions that are favorable to the use of a Gaussian plume model can be met.

# Acknowledgement

Funding for this research was provided by Climate-KIC. The authors thank Philippe Ciais and the industrial chair BridGES (Thales Alenia Space, Veolia, UVSQ, CEA, CNRS) for their support in this research. We would also like to acknowledge LSCE for the permission to conduct the measurements in the parking lot, particularly Pascal Doira for his availability and his help. Finally we thank the two anonymous reviewers, the associate editor D. Brunner, L. Golston and C. Herndon, J. R. Roscioli and T. Yacovitch for their detailed and technical comments that have helped strongly improve this study.

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

Table 1 − Weather conditions during the four tests and configuration of the observation vector for the statistical inversion.

| Trace gas configuration | Weather conditions (avg.) | | | Total number of transects | Number of selected transects | Configuration of the observation vector for the statistical inversion |
|---|---|---|---|---|---|---|
| | Temperature (°C) | Wind direction | Wind speed (m.s⁻¹) | | | |
| Configuration 1 | $9.9 \pm 0.3$ | N | $3.2 \pm 0.6$ | 29 | 11 | Integration of the entire plume |
| Configuration 2 | $9.2 \pm 0.1$ | N | $3.7 \pm 0.8$ | 20 | 9 | Integration of the entire plume |
| Configuration 3 | $8.4 \pm 0.8$ | N | $2.5 \pm 0.7$ | 35 | 10 | Integration of the entire plume |
| Configuration 4 | $11.3 \pm 0.3$ | NE | $2.0 \pm 0.7$ | 40 | 8 | Integration of slices of the plume |

Table 2 − Methane emission rates of the different controlled release configurations estimated with the different approaches and methane fluxes actually emitted during these tests. The uncertainties given with the tracer release method are detailed as follows: standard deviation of the random uncertainty derived from the variabilty of the results from one transect to the other one (bias due to the mislocation of the tracer ; total uncertainty).

| | Configuration 1 (collocated tracer) | Configuration 2 (upwind tracer) | Configuration 3 (lateral tracer) | Configuration 4 (multiple sources) |
|---|---|---|---|---|
| Controlled methane release (g.h⁻¹) | $382 \pm 7$ | $428 \pm 7$ | $360 \pm 7$ | $482 \pm 7$ |
| Tracer release method estimates (g.h⁻¹) | $434 \pm 23$ (0 ; 23) | $564 \pm 120$ (295 ; 415) | $321 \pm 51$ (43 ; 94) | $804 \pm 160$ (270 ; 430) |
| Relative difference to the control release (%) | 14 | 32 | 11 | 67 |
| Combined approach estimates (g.h⁻¹) | $441 \pm 6$ | $358 \pm 2$ | $386 \pm 2$ | $462 \pm 34$ |
| Relative difference to the control release (%) | 15 | 16 | 7 | 4 |

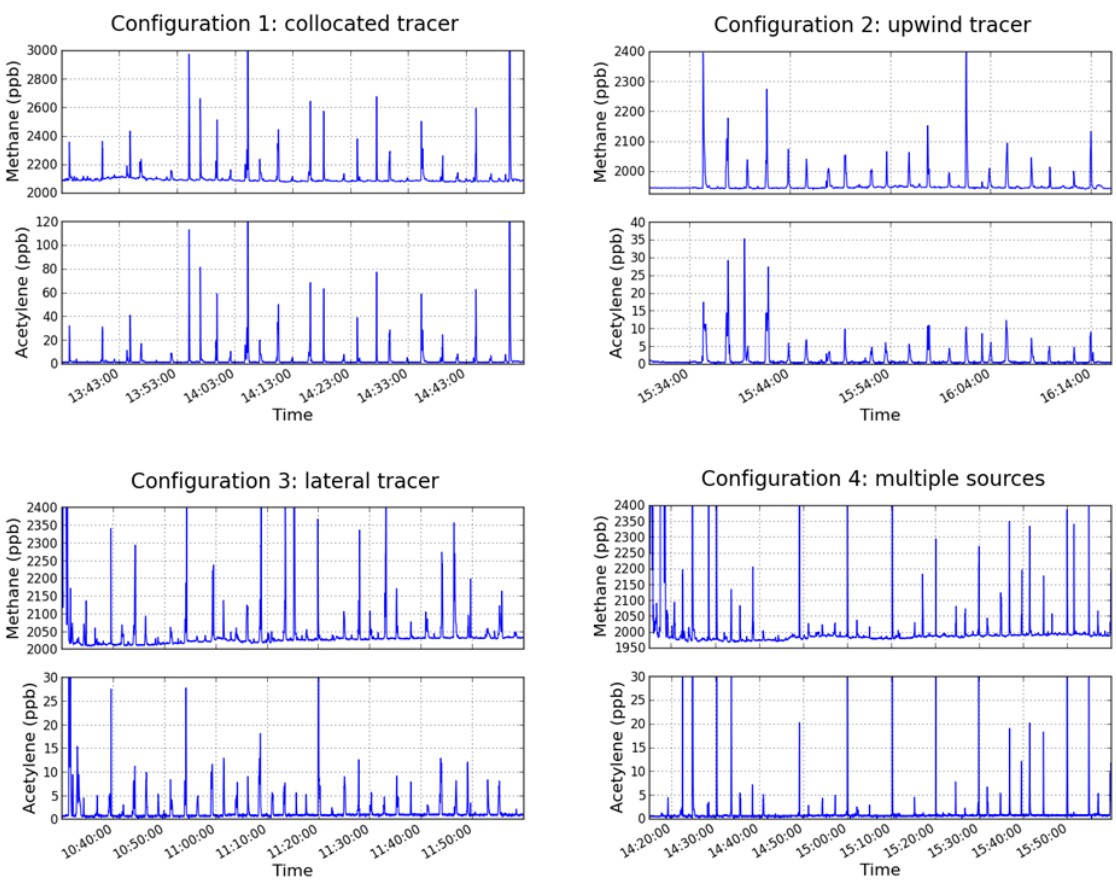

Figure 1 – Concentrations of methane and acetylene during the four tracer release experiments.

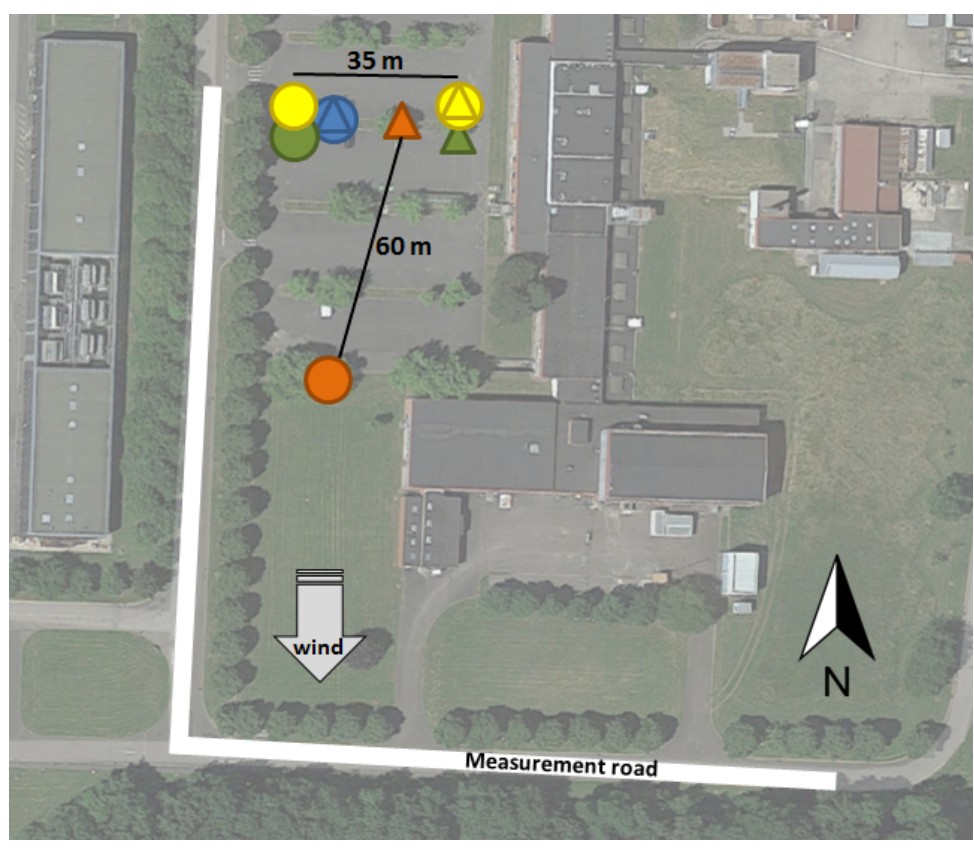

Figure 2 – The four tracer release configurations tested. Triangles represent the tracer source locations, and the circles mark methane sources. Each colour represents a configuration: blue is configuration 1 (collocated tracer), red is configuration 2 (upwind tracer), green is configuration 3 (lateral tracer) and, yellow is configuration 4 (multiple sources).

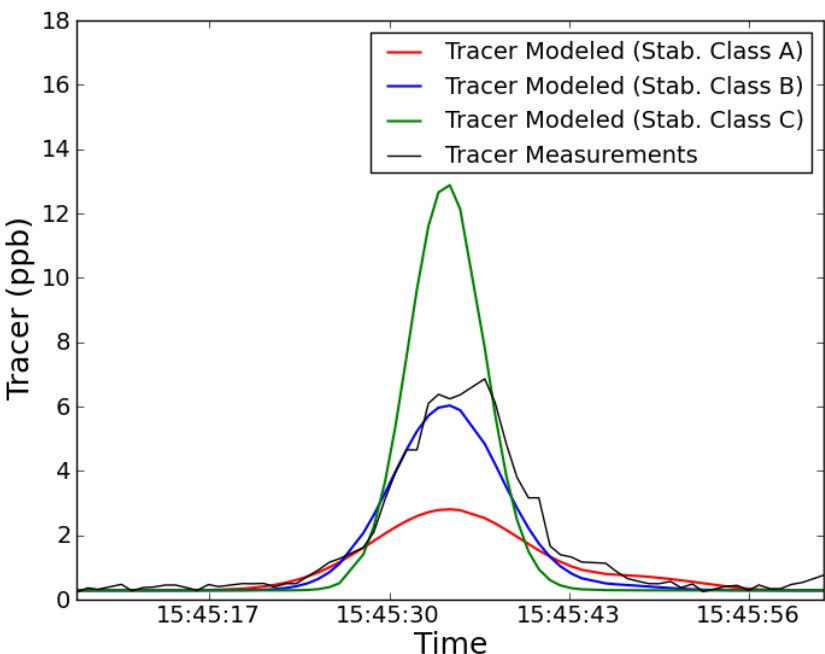

Figure 3 – Example of the Briggs parameterization selection with the acetylene data for peak 5 of configuration 2. The measured concentrations are presented in black, and the modelled concentrations with different stability classes are shown in colors.

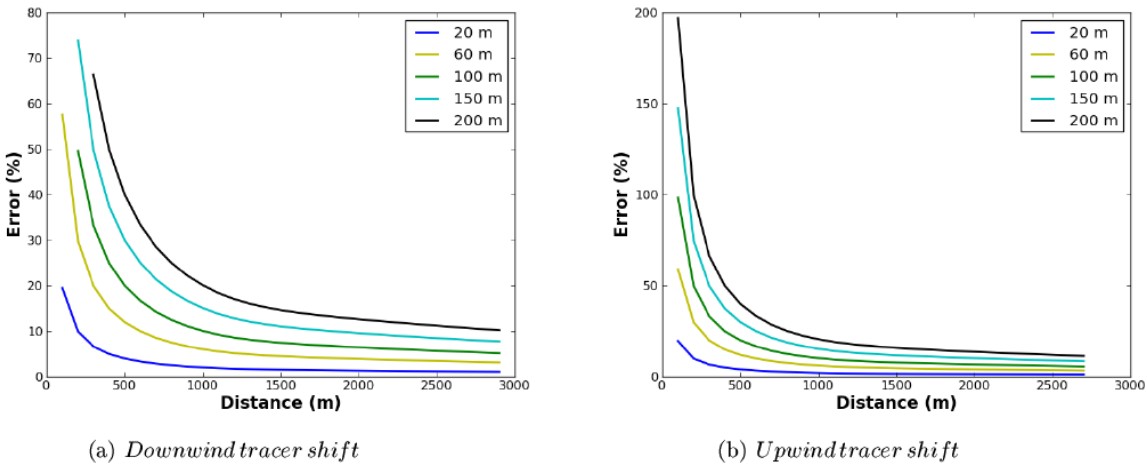

Figure 4 – Error in plume estimation with the tracer method depending on the measurement distance to the methane source and a shift of 20, 60, 100 150 and 200 m of the tracer location relative to the methane source using our Gaussian plume model.

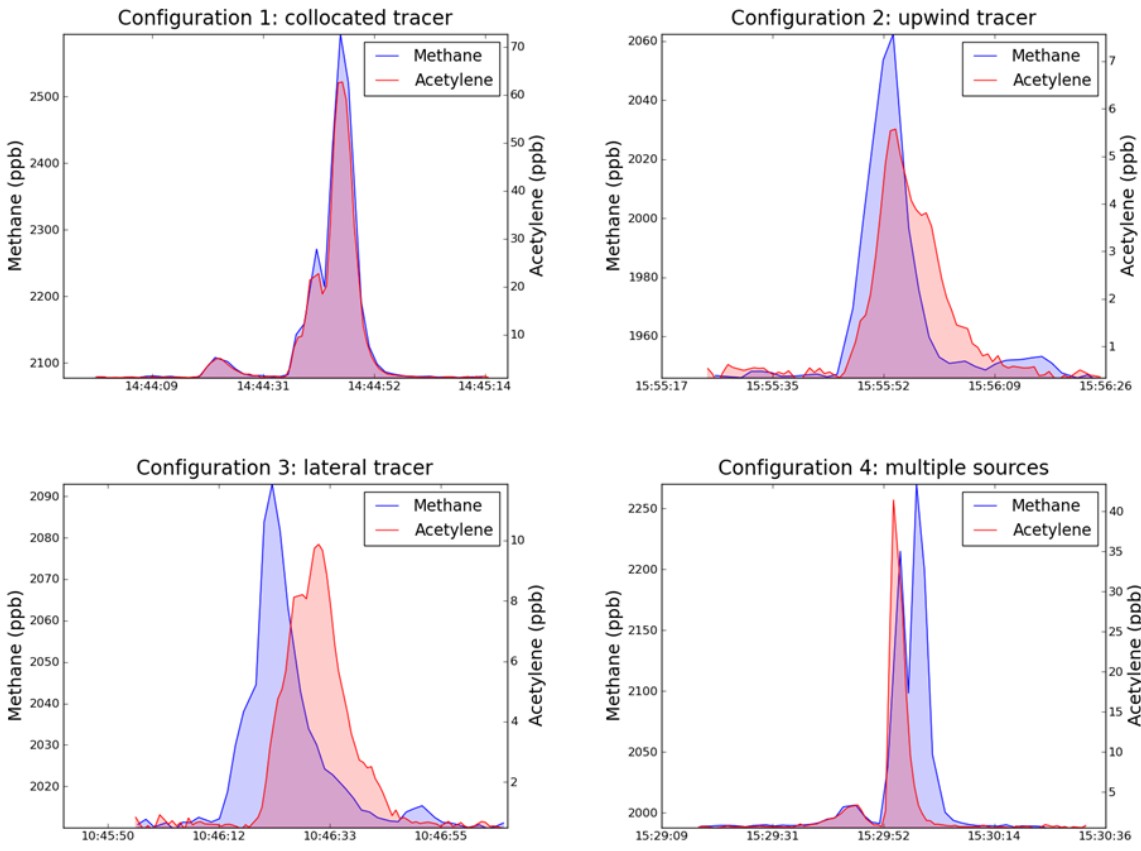

Figure 5 – Examples of cross-sections of the measured emission plumes of acetylene and methane (in red and blue, respectively) for each configuration.

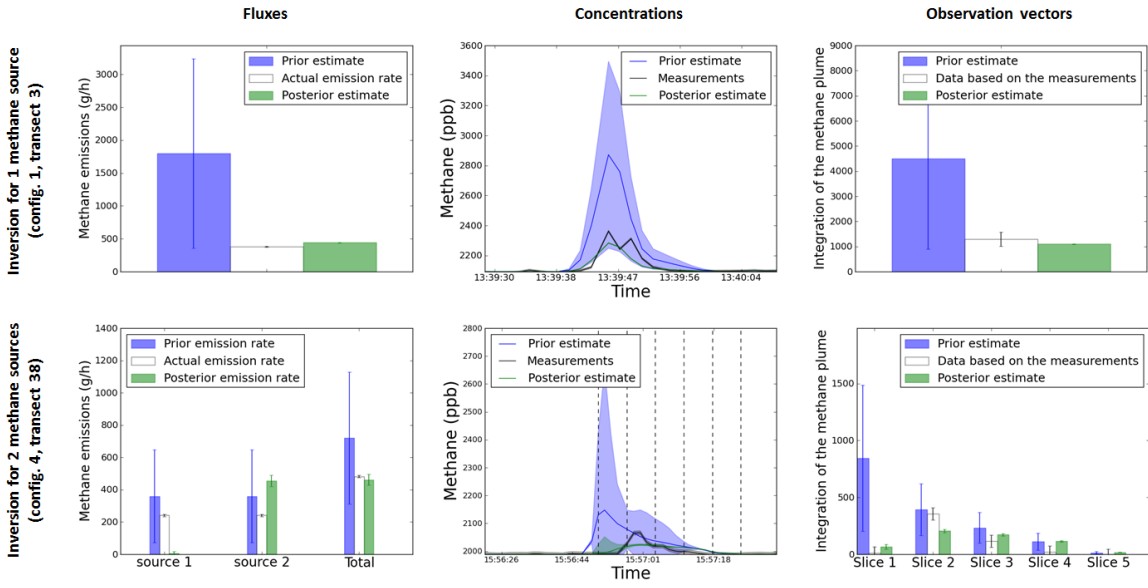

Figure 6 – Examples of prior, posterior and measured values of emission rates, concentrations and values of the observation vector for cases in configuration 1 and 4 (observations from a single transect shown).