# Peer review of "Statistical atmospheric inversion of local gas emissions by coupling the tracer release technique and local scale transport modelling: a test case with controlled methane emissions"

_Atmospheric Measurement Techniques, 2016_

## Referee Comment (RC1) · Anonymous Referee #1 · 12 Jan 2017

The authors propose a new method to estimate emission rates in industrial sites, combining the tracer release method, a Gaussian dispersion model and a statistical atmospheric inversion approach. The method is evaluated through field experiments and conclusions are drawn based on related results.

GENERAL COMMENTS. The idea of the authors is interesting, since it combines and merges different approaches. In their intention, this should bring benefit to the assessment and estimation of emission rates in complex industrial site. The scientific

approach is honest, by detailing all possible problems, and rigorous, by trying to address them. Indeed, the method is comprehensive and at the same time complex itself. As the authors discuss thoroughly in several parts of the manuscript, there are many degrees of freedom, many sources of possible uncertainties and errors, many unavoidable approximations, which can affect the reliability of the approach. For these reasons, I have not been fully convinced of the feasibility and applicability of the method. The conclusions are drawn from enough rigorous experimental tests but in a limited number and for a single site. Thus, it is not straightforward to infer whether the method can be effectively generalized, leading to a final novel procedure for estimating even unknown emission rates from industrial sites.

I have some specific concerns on the use of a Gaussian model. Certainly, for such short distances it can be reasonably applicable, as the author discuss, and its simplicity allows better dealing with the complexity of the problem. However, choosing a Gaussian model is questionable, since it cannot capture and describe turbulent motions, thus missing the fine-scale structures that can affect the results and their analysis. It has to be considered that turbulence and stochastic motions may produce uncertainties, for which also the tracer release method can fail, since turbulence acts altering the plume spread and pollutant dispersion. I think these aspects need to be better addressed and discussed in the manuscript, even if already several comments are spread in the text of this version, but too sparsely.

Regarding the manuscript and its form, the text is very dense, with a huge amount of information and comments, and avoidable repetitions. In some parts it gets strenuous to keep trace of the work done until first results are presented and finally discussed. The authors made an effort to provide a good organization of the paper structure, yet I think that it needs improvement, optimizing the description, removing repetitions, avoiding verbosity. The content and the form of the manuscript somehow are paired, in that a long and detailed description of the method and of the work done (almost 11 pages) then flow into confined results and conclusions (about 3.5 pages, plus tables

and figures). I think that the manuscript needs further revision before considering it for publication: more detailed comments are provided hereafter.

DETAILED COMMENTS.

* Introduction.

Page 2: why the locations of pollutant sources in industrial sites can be 'not always precisely known'? Because of possible 'fugitive emission' or leakages, or missing information from the industries? It will not be a matter of 'geo-localization', nowadays.

Page 2: 'local atmospheric dispersion models' might be of various type, from simple parametric methods to Gaussian, Eulerian, Lagrangian models. Since the authors cite 'models based on mass conservation' they should better specify that they are here referring to 'simple mathematical inversion' methods. In fact, advanced dispersion models are able to account for 'complex turbulence structures'.

Page 3: the authors state that the skill of statistical inversions approach strongly rely on the transport and source modelling. Then, a Gaussian model is used for the study. Gaussian models have strong limitations, especially when accuracy and turbulence structures are important factors, also given that they are designed for homogenous conditions. This is partly discussed later on, but some justification to support the use of a Gaussian models should be given here already.

* Section 2.2

Page 5: the authors explain their choice to minimize the impact of the differences between the targeted and released tracer plumes due to a not perfect collocation of the sources. Since turbulent motions can enhance the differences between the plumes further downwind the sources, is this the optimal choice for any distance from the emission points?

* Section 2.3

Page 5: I do not understand, and do not agree with the statement: "While LES and CFD models allow for turbulent patterns over such spatial scales to be generated and for changes in the terrain topography and for buildings to be accounted for (Letzel et al., 2008; Britter and Hanna, 2003), they can hardly be set-up or controlled to perfectly match the turbulent patterns at a given time and location downwind of a source." What do the authors mean? Why these models can be hardly set-up and controlled? Due to their complexity? No model can 'perfectly match' the turbulent patterns, but advanced models are in principle the best option, in particular when LES approach is used. Placed this way, this statement sounds just like a weak justification to use a simple Gaussian model, which on its side has instead severe limitations, since stationary solutions for homogeneous conditions are indeed a strong approximation of real atmospheric processes. Surely, advanced models (maybe available even in the Polyphemus system?) need more established modelling expertise and large computational resources for their application. It would be worth to include some discussion about the expected limitation when applying the Gaussian model in this specific site, where obstacles and buildings affect the flow and dispersion.

Page 6: Please, better explain in the text why "Instead of being deposited, the emission plume rebounds when it reaches the ground" is a 'decent' approximation for the studied gases.

* Section 2.5

Page 7: the new method intend to overcome the issues associated with the individual usage of the different methods. Could the author foresee possible 'new' uncertainties and issues linked or due to the merging of three approaches? I mean, a sort of propagation of uncertainties, of error propagation? Given the discussion about the limitations of the single method, I wonder whether it is proper to consider 'a priori' that the information on the atmospheric transport from the tracer release method and the Gaussian model simulations can be defined as 'very accurate information'. The final part of this subsection ("The statistics of the misfits...") is rather verbose, a bit compromising its

clarity. The multiple references to topics treated in next sections indicate possible repetitions. I suggest revising this part, optimizing the links with next ones, being more precise and less descriptive.

* Section 3.1

In which hours of the day were the measurements performed?

* Section 3.3

Page 8: please check the formalism accepted from the Journal for the units, if to use "l" for litre instead of "L", seconds "s" according to the SI instead of minutes "min" and hours "h".

* Section 3.5

Page 9: regarding the general applicability of the method, 'specific wind conditions of each cross-sections' to estimate the H observation operator are not commonly/routinely available, when not provided from experiments: could the authors comment on this aspect?

Page 10: some comments and interpretation about the best-fit obtained with stability class B (moderately unstable conditions) would be of interest: what were the atmospheric conditions during the experiment? Was B class effectively representative of them or the best agreement resulted 'by chance'? Here the authors 'admit' the limitation of the Gaussian model in reasonably reproducing the observed motion when turbulence, low wind etc may occur. So the question comes: why using a type of model that may not fit the purpose of its use? Also: how the threshold of 70% relative error was chosen to remove the data? Saying that the empirical choice has been defined based on the dataset is not enough, what was the reasoning behind?

Again, the cross-references to previous section 2.5 and 2.4 suggest that this part of the text should be optimized and harmonized with the previous one to avoid repetitions. Also, the description of the choices for the variances set-up might be improved, making

it less descriptive and clearer.

* Section 3.6

Minor: the title is rather long and descriptive, surely it is possible to shorten it. The authors may consider to combine this section with next section 4.1, since a few items are repeated and 3.6 is in fact functional to the results presented in 4.1.

* Section 4.1

See previous comment on section 3.6.

* Section 4.2

Page 13: it is redundant to repeat in the text the numbers already reported in the cited Table 2. Please revise this part to avoid such redundancy and repetition.

* Section 4.3

Same as for Section 4.2 about redundancy with Table 2. It would be worth to discuss more in depth why the tracer release method is better than the combined approach for the configuration 1. Because of fewer sources of uncertainties?

* Section 5.

In the results analysis and conclusions there is not a definitive and decisive proof that the combined method provides better and more reliable results than the tracer release method, given the limited number of cases and conditions considered, and the connected uncertainties. In addition, the potential ability of the method for multiple sources could not be fully addressed. The authors honestly recognize this and it becomes clear that these are 'preliminary' results and that more experiments are needed. Thus, the paper is mostly the presentation of a method but not a final test of it, supporting its adoption.

Figure 2: different colours (or line type) for the curves for different stabilities may better

[Figure]

highlight the results.

---

## Short Comment (SC1) · 21 Jan 2017

Ars et al. introduce a method for inverting small-scale emissions using a statistical framework which incorporates a Gaussian plume model and observations from the tracer method (here with acetylene). This is motivated because in real-world environments the exact location of the methane source could be spread out, inaccessible, or not precisely known, limiting the accuracy of the single tracer release technique by itself. Validation of the combined approach is attempted using a controlled methane and

acetylene release experiment in four different configurations.

I read this paper with interest since I am also doing work related to quantifying methane emissions at small scale. From the measurement side, the experiment appears to be well conceived and performed. However, I noticed multiple things that could be clarified or improved on the analysis side. I think there are some key revisions needed to helpful clarify the method and results, along with additional suggestions to improve the overall manuscript that are given in this comment.

Major comments

**1**. Physical basis: On the basis for the approach, it makes sense a the tracer releases could, for instance, constrain the dispersion in the Gaussian model when the tracer source is collocated with the methane source. However, when the tracer is positioned farther away, for all the reasons outlined in the paper (lack of a homogeneous / stationary atmosphere), the information from the tracer should become decreasingly useful as one has to rely more heavily on the model results. This is both because the diffusion of air along the acetylene path (obstructions, elevation changes, etc.) may not be the same, but also because a Gaussian model is used to bias correct for differences in downwind distance between methane and tracer. Yet, the test which performed the worst (in terms of relative difference) for the combined approach was actually Configuration 1, with a collocated tracer, while it performed better for all of the three conditions which should have increased the uncertainty of incorporating the tracer information.

It makes sense that the uncertainty would be higher for the combined approach than the tracer for Configuration 1, and that the tracer generally was worse for the non-collocated experiments. However, the result for the combined approach is unexpected, both from the perspective of the theoretical basis for the combined approach, and the interpretation of the actual measurement results. Does the combined approach account for uncertainty of the tracer technique when used under ideal (collocated) vs. non-ideal conditions? Why where the combined results better (again, in terms of

relative and absolute difference) when the tracer is used under less ideal conditions? More physical insight into how the combined approach works would be very helpful.

**2**. Organization: In general, there is unusually frequent referencing back and forth between Sections 2 and 3 and at one point even a "circular link" between section 2.5 and section 3.2 about the time lag, with neither quite containing the indicated information. Section 2 largely reviews the literature on these three techniques separate from the details of this paper, and could easily be condensed or even combined with section 3 since there are a lot of similarities between the two, and I think it would make it easier for the reader to understand specific aspects of the way the approach and experiment were conducted which is currently tedious going back and forth between the different sections and subsections.

**3**. Relationship to other literature: The effect of non-collocation, including distance of the measurement and magnitude of non-collocation, and the effect of being confined to the road which prevents non-orthogonal slices were discussed. These are all important issues for subsequent people using this method or similar experiments, and is also related to one of the conclusions [L676 - L679], so several recent papers would also be valuable to cite on these topics:

Goetz et al. 2015, Environ. Sci. Technol. (doi:10.1021/acs.est.5b00452) investigate issue where tracer not collocated and employs a correction based on the Gaussian plume

Roscioli et al. 2015, Atmos. Meas. Tech. (doi:10.5194/amt-8-2017-2015) also look at effect of the tracer and source not being collocated using the dual tracer framework to bracket possible errors, which is an alternative approach to what is given here and is likely applicable to similar types of sites

Albertson et al. 2016, Environ. Sci. Technol. (doi:10.1021/acs.est.5b05059) employs Bayesian framework and also specifically discusses and gives a correction for the issue of the road not being orthogonal to the wind direction also based on a Gaussian plume

formulation

**4**. Details missing that are important for understanding the approach/experiment:

- It is vague what information from the tracer is combined with the Gaussian, is it just the (rather coarse) adjustment of the stability class A-F, or more fine scale impact on the parameters ($\sigma$y, $\sigma$z, and/or wind)? It would be helpful to see how the parameters were actually affected during these experiments

- How the prior uncertainty is determined is not discussed, and the basis for model + observation uncertainty only briefly

- Both the method and results for the "multiple sources" inversion is brief other than that the plume is divided into five slices. Can a figure be added to illustrate how this works? Does using five slices mean up to five sources can be quantified? Can the approach resolve multiple sources when the plumes are overlapping, or only when they are basically non-overlapping?

**5**. Table 2: Two things stand out about Table 2, where results are given from the controlled release experiment.

First, why is there no row given for a Gaussian inversion, separate from combined approach? This is key information in evaluating the difference between the tracer release, Gaussian, and combined approaches.

Secondly, the uncertainty given for the combined approach is extremely small, several times smaller even than the controlled release uncertainty for Configurations 1, 2, and 3 which does not make sense. This is also noticeable in that for Configuration 2, the statistical chance of the uncertainty ranges 428 $\pm$ 7 and 464 $\pm$ 1 overlapping is 1-in-a-million, and for Configuration 1 the change of 382 $\pm$ 7 and 472 $\pm$ 2 overlapping essentially impossible. It is said all of the sources of error are considered, but this is clearly not the case - more thought should be given into how to derive a more representative uncertainty range for the combined approach.

**6**. Clarification on use of 'transient': The time dependence and transience of the problem is mentioned several times, but I do not think a clear explanation is given.

- In Figure 6 was the release not being run continuously? How was the time dependence of the prior derived?

- Additionally, the title says "Gaussian plume", which is formulated for a continuous, averaged value, not a transient release. Clarification is needed on this issue.

Minor comments

L487-491 and L556-559: Both good points

L492-495: Sentence could be clarified

Table 2: including the bias due to mislocation as part of the " +/- " does not make sense since it is not a random error. Also what about the uncertainty of the bias, since presumably this is non-trivial?

Table 1: one of the columns says 'wind direction (degree)', when that column is given in letters rather than degrees

---

## Referee Comment (RC2) · Anonymous Referee #2 · 9 Feb 2017

In Ars et al., the authors describe a new method for estimating gas emission rates from industrial facilities, by combining 1) tracer flux measurements, 2) Gaussian dispersion modelling and 3) a statistical inversion algorithm. The new method is evaluated using controlled methane/acetylene releases and compared to results from tracer flux. Four tracer placement scenarios are evaluated to demonstrate the improved accuracy of the method in situations where the tracer and the emission source are not perfectly collocated.

[Figure]

**GENERAL COMMENTS**

Overall, I was intrigued by the method presented by the authors, since accurately measuring industrial methane emissions remains a challenge, especially as we attempt to find convergence between bottom-up and top-down measurements. However, I am a little skeptical of value of the method in its current form and would like to see more discussion of the method's wider applicability before this paper is published in AMT.

Some areas of the method that I think warrant more discussion include the effect of different methane emission rates (only one was tested, $\sim$0.4 kg/h) and the role of meteorology – I am particularly concerned that the authors selectively looked at plumes from very specific atmospheric conditions. If this new method is being proposed as an "easy-to-implement" method for operators to employ (as it is described in the Introduction), then I would expect such a method to be robust to different atmospheric stabilities. The quality of the writing is excellent, but the authors would do well to streamline the paper so it is less bogged down in text.

I tend to agree with Reviewer #1 who described the writing as "verbose". This complex writing style makes it more challenging to follow the science. Additionally, I think the authors could limit some of the discussion of the methods, particularly the tracer release and Gaussian methods, as these are well-described in the literature, to make room for a more well-rounded discussion of the results, which seemed rushed. Upon making these major revisions, I expect the publication will be suitable for publication in AMT. Specific comments follow.

**SPECIFIC COMMENTS**

Section 1 – L105: Is this method easy to implement for operators? How have operators historically monitored their emissions? If the paper is framed as being in support of industry, then this should be discussed; I am not familiar with many facilities actively conducting tracer flux measurements or those with mobile laboratories to measure downwind emissions.

Section 2.1 – L175: I am not sure how you have demonstrated that your method provides satisfying results over those distances and methane emission rates compared to what you have tested with your controlled releases – please elaborate. I am especially interested in how tracer flux and this method differ for large methane emission rates or "superemitters".

Section 2.2 – L180: I suggest that the authors conduct a more thorough literature review, particularly of tracer release measurements conducted in various shale gas basins in the United States. Numerous papers have come out on this subject in the past 3 years.

Section 2.2 – L210-215: Can the authors speak to how this effect scales with methane emission rate? Does its significance shrink if total methane emissions increase? Or does its importance scale linearly?

Section 2.3 L230-235: Can you expand on this more in the text? I find the model justification to be a little lacking.

Section 2.3 L250-253: Is this detail on urban vs. rural configurations really necessary? Especially if you don't mention what configuration was used in this study.

Section 2.5 L305-310: I went looking for an explanation of how the spatial offset was treated in Section 3.2, but this section reference Section 2.5. Please make sure this concept is explained. I would strongly caution against routinely referencing other sections, particularly future sections, and instead focus on a linear narrative for the paper.

Section 3.1 L342-345: If the authors are going to be highly selective of meteorological conditions, then this should be discussed in more detail. What happens on days with low winds?

Section 3.2 L360-364: Here are some more cyclical references – I don't think the spatial offset is ever properly described.

Section 3.5 L430-435: Choosing stability class based on best fit to the measurements

seems suspect to me. How does this choice compare to the estimated stability class using wind speed and insolation metrics? Furthermore, what were the range of atmospheric stabilities during all your tests? Is this method applicable to all stability classes? This is a main point of concern for me and the authors should better justify their decision regarding the Briggs parameterization.

Section 3.5 L472-479: I could not follow this; can you explain how these uncertainties translate into those methane emission rates?

Section 4.2 L584-590: I am not convinced by the argument that the performance of the new model was the worst compared to the actual emission rates due to the low emission source – it seems to me all the other configurations used comparably low emission rates and this problem was not observed. Please provide a better explanation.

Section 4.2 L584-613: This is repetitive of table 2 and does not to be listed off in the text.

Section 4.3 L628-652: Again, this is repetitive, I would prefer to see more of an analysis vs. repeating of figures in a Table.

Section 4.3 L432: The authors explain the poor performance in configuration 1 does not matter very much due to the fact that the configuration is unrealistic. I am not satisfied with this explanation, if theory dictates that unreasonable or not that configuration 1 should be the ideal case then a good reason should be provided why it was not.

Section 5: Nowhere in the conclusions (or in the results) do I see any statements on the performance of this method vs tracer flux for a range of methane emissions. This was introduced in the introduction and I do not think it was adequately followed through on. If this method is currently limited to low industrial emission rates it should be expressly stated. As it stands, I think the usefulness of the method is overstated and the authors should be realistic about what their experiments have demonstrated.

Table 1: If I understand correctly, on the days where meteorological conditions were

explicitly controlled for, the plume capture rate is roughly 30%. This seems very low to me making me question the robustness of the method. Please comment.

TECHNICAL CORRECTIONS

Section 2.3, L219: Please define acronym "LES" and possibly "CFD", as I am unsure if everyone would know what these are.

Section 3.1 L340: Typographical error "serie"

Section 3.6: Edit section title to be more succinct.
* * *

---

## Short Comment (SC2) · 14 Feb 2017

Review of

"Statistical atmospheric inversion of small-scale gas emissions by coupling the tracer release technique and Gaussian plume modeling: a test case with controlled methane emissions"

by S. Ars et al.

[Figure]

Submitted by Scott C. Herndon, Joseph R. Roscioli and Tara I. Yacovitch, Aerodyne Research, Inc.

General Comments:

Overall, we think this is a rigorous methods paper and does a good job of describing a novel combination of tracer and dispersion methodologies. The descriptions of the experiments and methods are clear, and will be well understood by the community, both on the experimental side and on the dispersion side. We think this paper should be published after addressing the comments below.

1. Please cite some important related research that explores tracer mislocation effects:

Goetz et al. used a similar combined methodology in 2015 for tracer release experiments at wellpads. Goetz, J. D.; Floerchinger, C.; Fortner, E. C.; Wormhoudt, J.; Massoli, P.; Knighton, W. B.; Herndon, S. C.; Kolb, C. E.; Knipping, E.; Shaw, S. L.; et al. Atmospheric Emission Characterization of Marcellus Shale Natural Gas Development Sites. Environ. Sci. Technol. 2015, 49, (11), 7012; DOI: 10.1021/acs.est.5b00452.

Roscioli et al. Performed an extensive error analysis of the impact of tracer mislocation using dual-tracer release methodology. Additional methods of calculating the methane/tracer ratio are also described. Roscioli, J. R.; Yacovitch, T. I.; Floerchinger, C.; Mitchell, A. L.; Tkacik, D. S.; Subramanian, R.; Martinez, D. M.; Vaughn, T. L.; Williams, L.; Zimmerle, D.; et al. Measurements of methane emissions from natural gas gathering facilities and processing plants: measurement methods. Atmos. Meas. Tech. 2015, 8, (5), 2017; DOI: 10.5194/amt-8-2017-2015.

2. Please address some potential problems with the experimental measurements: Have experimentally measured mixing ratios been calibrated? Please briefly mention calibration procedure.

Line 370-375: The flow meter you refer to is an analog measurement of flow rate, so we don't think you have a time-resolved record of the flow rates. While you have a final

check of the release by mass difference, do you have any estimate of the variability of the flow rate over the course of the experimental measurements? This could be substantial, particularly with acetylene releases, which can vary as the cylinder cools. In such a case, the flow rate may appear the same on the flow meter, but the actual mass flow may be different.

Paragraph starting at 441: When multiple instruments are sampling in a mobile vehicle, they will often have different inlet times (lag due to air intake). This can be corrected based on experimental measurements, for example by delivering an excess of nitrogen or zero-air to the inlet tip, and monitoring the instrument responses. Has this been done experimentally? How does the time-shifting of data based on simulated plumes compare to the experimental lags?

3. Why are ideal tracer ratio experiments not producing acceptable results?

Config. 1 tracer release method estimates before Gaussian correction (e.g. Table 2, middle row) overestimate the true release rate by 19%, when it should produce accurate results without the need for correction (see discussion below for more detail). This discrepancy casts doubt on the biases of the remaining 3 configuration tests. The calibration of mixing ratios and the consideration of errors in the instantaneous flow rates (see above) should be considered.

Lines 584 – 590: Something seems wrong with this plume analysis. You are well above the detection limit (at least looking at Figures 4 and 5), so the explanation of why the tracer analysis is not accurate does not make sense. Configuration 1 should be the ideal case. Please rule out instrument calibration errors (both in measurement and release equipment), unit errors (temperature and pressure of measurement), and other experimental issues.

Dual-tracer release has been used in the past to quantify bias. When they are collocated, the emission rate of one tracer as derived from its downwind ratio to the other is found to agree extremely well with the known mass flow (much better than the 19%

error that is observed here). See Figure S4-4 of the supporting information for Allen, et al, Proc. Nat. Acad. Sci. (PNAS), vol 110, page 17768 for an example. It is also shown in the attached figure. There, the downwind ratio of two collocated tracers ($C_2H_2$ and $N_2O$) is within 1% of the ratio of their mass flow rates (Figure S4-4c). If $N_2O$ were replaced by $CH_4$, the scenario in Figure S4-4 would be experimentally identical to configuration 1 of this manuscript. The tracer release-derived $CH_4$ flow rate would then be within 1% of the known flow rate. This level of agreement for collocated sources is routine in the field. Therefore, the 19% deviation observed here strongly suggests that there is an issue with the measurement method (instrumental), or tracer/methane flow rate.

If configuration 1 were viewed a "control", then the +19% bias indicates that other aspects of the measurements are +19% biased, not the tracer method inherently. In that case, it should be used as an bias offset – that is, the +29% bias for configuration 2 should actually be 29%-19% = 10%, for configuration 3 should be 17%-19% = -2%, and for configuration 4 should be 58%-19% = 39%.

Lines 605 – 613: Please better explain why this tracer ratio method is not working, given that earlier you say that position errors perpendicular to the wind should not have such a large effect. Was the wind varying? Also, the "configuration 4" panel in Figure 5 shows only one methane plume. Given that there are two $CH_4$ sources, why don't you see two $CH_4$ plumes, or at least a broad $CH_4$ plume? We would expect the two $CH_4$ plumes to look like a composite of the $C_2H_2$ and $CH_4$ plumes in configuration 3, which has identical separation.

4. Initial guesses for optimization

Paragraph at 276 & Line 474: Please explain how initial guesses completely independent of measurements can be done in practice, particularly for emission sources where the expected emission magnitudes may span many orders of magnitude (e.g. factor of 100, instead of 80%). This would be the case, for example, for certain oil and gas

emission experiments. Is it possible to use tracer release result (without any dispersion corrections) as the prior?

Specific Comments:

Lines 191 - 196: Other methods of calculating the ratio are commonly used, notably taking the slope of the plot of the methane vs tracer plume signals. This is generally found to be more precise than measuring the area under each plume, because it does not depend upon choice of background. See Roscioli et al: Roscioli, J. R.; Yacovitch, T. I.; Floerchinger, C.; Mitchell, A. L.; Tkacik, D. S.; Subramanian, R.; Martinez, D. M.; Vaughn, T. L.; Williams, L.; Zimmerle, D.; et al. Measurements of methane emissions from natural gas gathering facilities and processing plants: measurement methods. Atmos. Meas. Tech. 2015, 8, (5), 2017; DOI: 10.5194/amt-8-2017-2015.

Lines 225 : There is considerable debate in the atmospheric modeling community on the timescale appropriate for the canonical A-D stability classes (e.g. Pasquill-Gifford stability classes). The consensus seems to be more on the order of 10-15 minutes than 1-2 minutes. See, for example: Fritz, B. K.; Shaw, B. W.; Parnell, C. B. J. Influence of meteorological time frame and variation on horizontal dispersion coefficients in Gaussian dispersion modeling. Trans. ASABE. 2005, 48, (3), 1185; DOI: 10.13031/2013.18501

Line 302: "... spatial offset between the measured plume and the actual plumes due to the lag between the air intake and the concentration measurement." I recommend adding the parenthetical statement: (also known as inlet lag or inlet time)

Line 340: correct "serie" to "series"

Line 351: While the instrument reporting time is noted (2 seconds), the instrument response time is not. Furthermore, the data depicted for configuration 1 in Figure 5 suggests the time response for the two channels is not fully matched. Is the instrument reporting all mixing ratios simultaneously, or does the instrument sub-divide the

2 second interval to quantify each of the noted species.

Line 429: Is the 5th percentile of transect concentrations sufficient to determine baseline? Do the results change if the 2nd percentile, or 10th percentile is used?

Lines 561 to 567 and Figure 3: Why not show these results as a function of relative distance downwind, i.e. (distance between source and tracer)/(distance between site and measurement)? The distances in meters shown can not be easily generalized. I further suggest putting panels a) and b) on the same vertical scale, and adding gridlines at the same intervals for both.

All Figures: Please increase font size of all labels and numbers so that they are readable at the width of a normal sheet of paper.

Figure 4: Consider showing only Figure 5 (representative plume transects) instead of the whole data set. These full data results, depicted in Figure 4, might be better left to the supplementary info. Even better, consider publishing these results as a test dataset. If Figure 4 is to remain, then the vertical axes should be rescaled to see all of the plume intensities.

Figure 6: Figure 6 is the most important figure of the manuscript and needs to be reformatted for legibility. It is currently much too small. Consider perhaps a small cartoon drawing of "config. 1" and "config 4". To clarify the difference between the 2 sets of results shown. Please also label the meaning of the shaded area ("concentrations" plots) in the legend or figure title.

Line 565: Instruments with lower levels of detection than the one used here are available. Please alter this statement to reflect this fact: e.g.: "...signal to measurement noise ratio would likely be too small, using these instruments, to derive..."

Section 4.3, Lines 614: Please reference Goetz et al, who in 2015 used a similar approach to this.

Line 651: It is possible, (see Roscioli et al.) in some cases, to co-locate tracers and

emission vectors for real experiments. Please rephrase "which can hardly occur"

[Figure]

[Figure]

Figure S4-4 a.) Methane, acetylene and $N_2O$ plumes observed downwind of a production site; tracers were co-located; b.) the average ratio of methane to $N_2O$ in the plume, determined using second by second observations of methane and $N_2O$ is indicated by the slope of the line; this ratio was used in Equation S4-2, with the known release rate for $N_2O$, to estimate methane emissions; c.) the average ratio of acetylene to $N_2O$ in the plume determined using second by second observations of acetylene and $N_2O$ in the plume is indicated by the slope of the line; the 0.8% error indicates the difference between the ratios of the observed concentrations in the plume and the ratios of the tracer release rates for this site.

**Fig. 1.**

---

## Author Comment (AC1) · 14 Jun 2017

The authors propose a new method to estimate emission rates in industrial sites, combining the tracer release method, a Gaussian dispersion model and a statistical atmospheric inversion approach. The method is evaluated through field experiments and conclusions are drawn based on related results.

GENERAL COMMENTS. The idea of the authors is interesting, since it combines and merges different approaches. In their intention, this should bring benefit to the assessment and estimation of emission rates in complex industrial site. The scientific approach is honest, by detailing all possible problems, and rigorous, by trying to address them. Indeed, the method is comprehensive and at the same time complex itself.

We thank the reviewer for this general assessment and for his constructive, detailed and technical comments that will help strongly improve our manuscript, and in particular the presentation of the objectives, concepts, and long-term perspectives of our study.

As the authors discuss thoroughly in several parts of the manuscript, there are many degrees of freedom, many sources of possible uncertainties and errors, many unavoidable approximations, which can affect the reliability of the approach. For these reasons, I have not been fully convinced of the feasibility and applicability of the method. The conclusions are drawn from enough rigorous experimental tests but in a limited number and for a single site. Thus, it is not straightforward to infer whether the method can be effectively generalized, leading to a final novel procedure for estimating even unknown emission rates from industrial sites.

We assume that the success of this method, and, at least, its ability to provide better results than the traditional tracer release technique will depend on the configuration (size and extent of the sources, topography, local transport conditions, positioning of the roads…) of the industrial sites to be investigated. But, at least, this paper demonstrates, both theoretically and for a practical case, that it in principle, it has some potential to behave better than the tracer release technique (which is used quite extensively for the estimate of industrial sources nowadays: Roscioli et al. 2015, Goetz et al. 2015, Taylor et al. 2016). We think that this is definitely worth leading to new studies and tests of such a concept and analyzing how to improve and generalize it. Our OSSEs (see section 3.6) demonstrate severe issues when using the tracer release technique even for simple point source estimation cases, while the use of this method is definitely generalized, highlighting the need for exploring new techniques. The proposed method is new, we naturally acknowledge the need for improving it (see the point regarding the transport model below) in order to limit the impact of the sources of uncertainties that we have discussed.

We will emphasize these points in the introduction and in the conclusion. Of note is that the conclusion will be expanded into a sort of discussion/conclusion section to gather many of the digressions that lengthened the previous sections and to include discussions asked by the reviewers.

I have some specific concerns on the use of a Gaussian model. Certainly, for such short distances it can be reasonably applicable, as the author discuss, and its simplicity allows better dealing with the complexity of the problem. However, choosing a Gaussian model is questionable, since it cannot capture and describe turbulent motions, thus missing the fine-scale structures that can affect the results and their analysis. It has to be considered that turbulence and stochastic motions may produce uncertainties, for which also the tracer release method can fail, since turbulence acts altering the plume spread and pollutant dispersion. I think these aspects need to be better ad-

**dressed and discussed in the manuscript, even if already several comments are spread in the text of this version, but too sparsely.**

We will expand the discussions on these aspects.

Of note, regarding the use of the Gaussian model in this study:

First, by jointly assimilating the data from the different transects through the methane plumes in the inversion, we attempt at imposing a sort of "average plume" as the main constraint on the emissions. This should limit the weight of fine-scale structures.

Second, the lack of turbulent structure in the Gaussian simulation is implicitly accounted for when assessing the model error based on C2H2 model vs. data comparisons. So the uncertainty in the inverted emissions associated with the lack of turbulent structures is included in the diagnostic of uncertainty by the inversion system.

At last, the success of our method with a Gaussian model compared to the tracer release method in our practical test demonstrates the relevance of using such a model for the first assessment of our inversion concept.

Models more sophisticated than Gaussian models could produce turbulent motions but one would hardly manage to control them for capturing (in the sense of having them at right time and location) that seen in the data especially with the available dataset from the type of measurement campaigns we consider. The reviewer criticizes this assumption (see comment on 2.3 and our answer to this comment) and we are ready to be convinced by his considerations on this. Furthermore, using more complex models would allow to account for complex topography, for temporally and spatially varying local meteorological conditions… We thus now prefer to emphasize that testing our theoretical concept first with a model as simple as a Gaussian model before increasing the complexity of the problem made sense since our results demonstrate some advantages of this modeling framework compared to the tracer release technique.

These points will be better emphasized in the manuscript. Furthermore, section 2.3 (which will be split into 2 subsections: a general but short one on local scale transport models and a more specific one on Gaussian models) and the conclusion will acknowledge that some improvement of the method could be achieved by using models more complex than Gaussian models, at least for applications in places with complex topography or situations with complex meteorological conditions, even though we feel that exploiting the capability of such models to generate turbulent patterns would not be straightforward.

**Regarding the manuscript and its form, the text is very dense, with a huge amount of information and comments, and avoidable repetitions. In some parts it gets strenuous to keep trace of the work done until first results are presented and finally discussed. The authors made an effort to provide a good organization of the paper structure, yet I think that it needs improvement, optimizing the description, removing repetitions, avoiding verbosity. The content and the form of the manuscript somehow are paired, in that a long and detailed description of the method and of the work done (almost 11 pages) then flow into confined results and conclusions (about 3.5 pages, plus tables and figures).**

We will try to use more subsections, use references to such subsections to avoid redundancies and more generally try to be more concise. We want to keep the general structure as it is and gather sections 3 and 4 into a single section 3  since we feel that this paper has definitely two components: the definition of a new theoretical framework in one hand (the present section 2), and its evaluation through an academic experiment (the present sections 3 and 4). It is thus difficult to compare the length of the present section 4

to that of the present sections 2 and 3.

**I think that the manuscript needs further revision before considering it for publication: more detailed comments are provided hereafter.**

We hope that our answers will provide convincing indications of the improvement we plan for the paper.

**DETAILED COMMENTS.**

**\* Introduction.**

**Page 2: why the locations of pollutant sources in industrial sites can be 'not always precisely known'? Because of possible 'fugitive emission' or leakages, or missing information from the industries? It will not be a matter of 'geo-localization', nowadays.**

Fugitive emissions (like leakages) that are transitory, or affecting poorly reachable areas or complex buildings, unexpected sources, but also widespread and heterogeneous sources (e.g. livestock in the building of a farm, basins in waste water treatment plants or cells in landfills for which emissions are not homogeneously distributed) makes it difficult to know perfectly the distribution of the emissions within a site. This sentence will be modified and extended to clarify it. In particular we will change "location of pollutant sources" into "spatial distribution of the emissions".

**Page 2: 'local atmospheric dispersion models' might be of various type, from simple parametric methods to Gaussian, Eulerian, Lagrangian models. Since the authors cite 'models based on mass conservation' they should better specify that they are here referring to 'simple mathematical inversion' methods. In fact, advanced dispersion models are able to account for 'complex turbulence structures'.**

We will improve this introduction on the local dispersion models mentioning families of available models (CFD/LES, Lagrangien…).

**Page 3: the authors state that the skill of statistical inversions approach strongly rely on the transport and source modelling. Then, a Gaussian model is used for the study. Gaussian models have strong limitations, especially when accuracy and turbulence structures are important factors, also given that they are designed for homogenous conditions. This is partly discussed later on, but some justification to support the use of a Gaussian model should be given here already.**

See our general answer to the similar general comment above regarding the turbulent patterns. We will also try to better discuss the limitations of the method when using a Gaussian model, which could limit its applicability to simple cases in terms of topography and meteorology. The conclusion will remind that, in principle, the general concept of the inversion could be used with other types of models for the sake of improvement and / or generalization, but that the details of such an implementation would still have to be studied.

**\* Section 2.2**

**Page 5: the authors explain their choice to minimize the impact of the differences between the targeted and released tracer plumes due to a not perfect collocation of the sources. Since turbulent motions can enhance the differences between the plumes further downwind the sources, is this the optimal choice for any distance from the emission points?**

We will better indicate that part of the impact due to a not perfect collocation of the sources can be

emphasized by incompatible turbulent patterns between the tracer and targeted species. We assume that the author questions our choice of using the area below the plume instead of its maximum to characterize the plume. In principle, whatever the distance from the source, there is more chances that the maximum of the plume is more impacted than its area by the turbulent patterns than the opposite. But it definitely depends on the situation (on the structure of the plume). We have not thought about more sophisticated diagnostics of the plume that would be less sensitive to these turbulent patterns.

**\* Section 2.3**

**Page 5: I do not understand, and do not agree with the statement: "While LES and CFD models allow for turbulent patterns over such spatial scales to be generated and for changes in the terrain topography and for buildings to be accounted for (Letzel et al., 2008; Britter and Hanna, 2003), they can hardly be set-up or controlled to perfectly match the turbulent patterns at a given time and location downwind of a source." What do the authors mean? Why these models can be hardly set-up and controlled? Due to their complexity? No model can 'perfectly match' the turbulent patterns, but advanced models are in principle the best option, in particular when LES approach is used.**

We agree that advanced models are in principle the best options, but we mean that their advantages do not seem easy to exploit for such an inversion problem. Our sentence is not in opposition with this but we will try to clarify and moderate it. We do not mean that these models are difficult to set-up, but that it is not straightforward to control them to ensure that their turbulent patterns are placed approximately well, a requirement to take advantage of it in inversions that make a simple minimization of simulated vs. measured data at different times and locations. Gaussian models seem a good choice for the first test of our method considering their very low calculation costs and the simplicity of their application.

**Placed this way, this statement sounds just like a weak justification to use a simple Gaussian model, which on its side has instead severe limitations, since stationary solutions for homogeneous conditions are indeed a strong approximation of real atmospheric processes**.

We will now better clarify in the introduction and conclusion that the use of a Gaussian model can be seen as a first step to test our theoretical framework with a simple model before increasing the level of complexity. The successes obtained with this model in our practical case demonstrate that this was relevant. And again, the conclusion will now indicate that future studies should investigate the use of more sophisticated models and the appropriate strategies to take benefits from their advantages for the sake of improvement and generalization of the method to site configurations with complex topography and local meteorological conditions.

**Surely, advanced models (maybe available even in the Polyphemus system?) need more established modelling expertise and large computational resources for their application. It would be worth to include some discussion about the expected limitation when applying the Gaussian model in this specific site, where obstacles and buildings affect the flow and dispersion.**

We will do it.

**Page 6: Please, better explain in the text why "Instead of being deposited, the emission plume rebounds when it reaches the ground" is a 'decent' approximation for the studied gases.**

We will replace "Instead of being deposited, the emission plume rebounds when it reaches the ground is a 'decent' approximation for the studied gases" by "As both studied gases are poorly soluble and chemically inert for the dispersion time scale we consider in this study it is relevant to neglect the mass loss due to dry deposition and assume a total reflection from the ground."

**\* Section 2.5**

**Page 7: the new method intends to overcome the issues associated with the individual usage of the different methods. Could the author foresee possible 'new' uncertainties and issues linked or due to the merging of three approaches? I mean, a sort of propagation of uncertainties, of error propagation? Given the discussion about the limitations of the single method, I wonder whether it is proper to consider 'a priori' that the information on the atmospheric transport from the tracer release method and the Gaussian model simulations can be defined as 'very accurate information'.**

We will better demonstrate the moderation of our expectations from this method and its possible limitations. The model uncertainties are explicitly quantified and accounted for in the inversion system. The estimate of prior and model uncertainties which are required by the statistical inversion are definitely not straightforward even though we have data to support the derivation of the latter. Still, weighting such uncertainties in the estimate of emissions is safer than just ignoring them in "deterministic inversion" (based on the tracer or a transport model).

**The final part of this subsection ("The statistics of the misfits. . .") is rather verbose, a bit compromising its clarity. The multiple references to topics treated in next sections indicate possible repetitions. I suggest revising this part, optimizing the links with next ones, being more precise and less descriptive.**

Following this comment, we will rework the different parts to prevent these references and be more concise.

**\* Section 3.1 In which hours of the day were the measurements performed?**

Measurements hours are detailed in figure 4, we will give this information more specifically in the section 3.1.

**\* Section 3.3**

**Page 8: please check the formalism accepted from the Journal for the units, if to use "l" for litre instead of "L", seconds "s" according to the SI instead of minutes "min" and hours "h".**

We have not found the proper formalism of the journal for these units, but it should be corrected during the final steps of the editing process.

**\* Section 3.5**

**Page 9: regarding the general applicability of the method, 'specific wind conditions of each cross-sections' to estimate the H observation operator are not commonly/routinely available, when not provided from experiments: could the authors comment on this aspect?**

We will indicate that the corresponding meteorological variables, if not measured during the experiments, are hardly accessible from local meteorological networks. Analyses by operational centers are available but their spatial representativeness is not always adapted to the characterization of local conditions. We thus strongly support conducting meteorological measurements along with the gas measurements if willing to use a modeling framework as here.

**Page 10: some comments and interpretation about the best-fit obtained with stability class B (moderately unstable conditions) would be of interest: what were the atmospheric conditions during the experiment? Was B class effectively representative of them or the best agreement resulted 'by**

**chance'?**

The measured wind speed during this series was 3.7 m/s. For this wind speed, the Pasquill classification indicates that the corresponding stability is whether B whether C depending on the solar radiation. The solar radiation information may be difficult to have, especially at this scale, and it is thus difficult to choose between these two stability classes without the tracer data. Therefore, in this example and for all transect, we have checked that the selected stability class is part of those that are in agreement with what we know of the meteorological conditions.

We will better discuss it in this section.

**Here the authors 'admit' the limitation of the Gaussian model in reasonably reproducing the observed motion when turbulence, low wind etc may occur. So the question comes: why using a type of model that may not fit the purpose of its use?**

The main purpose is to catch some information about the emissions characterized by the "mean plume" of methane rather than by the turbulent patterns on the top of it. Figure 2 indicates that once its stability class is optimized, the model fits very well with the 1$^{st}$ order pattern from the tracer source which is our aim so we do not agree that our model does not fit the purpose of its use. Again, in a more general way, the success of the method compared to the classical tracer release method for our specific test case demonstrate that the Gaussian model already provides a relevant skill for our objective. But, again, we now indicate in conclusion that more complex models could help improve the accuracy of the estimation and generalize the applicability of the method.

**Also: how the threshold of 70% relative error was chosen to remove the data? Saying that the empirical choice has been defined based on the dataset is not enough, what was the reasoning behind?**

This threshold fits with our qualitative analysis of the comparisons between measured and simulated tracer concentrations since it removes the transects for which we consider that the shape of the measured and simulated tracer plumes are quite different (simulated transect being wider than measured transects in most of these case) whereas for the other transects the Gaussian model managed to represent quiet well the measured profiles. Still, this conservative choice is not critical since the model error associated with each transect in the atmospheric inversion is based on the fit between the modeled and simulated tracer plumes, so that transects for which this fit is rather low do not bring a significant constraint on the inversion results.

**Again, the cross-references to previous section 2.5 and 2.4 suggest that this part of the text should be optimized and harmonized with the previous one to avoid repetitions. Also, the description of the choices for the variances set-up might be improved, making it less descriptive and clearer.**

We will make sections 2.4 and 2.5 shorter by detailing the specific concepts for the adjustment of the model data comparisons mostly in section 2.5 and we will improve the presentation of the variances.

**\* Section 3.6**

**Minor: the title is rather long and descriptive, surely it is possible to shorten it. The authors may consider to combine this section with next section 4.1, since a few items are repeated and 3.6 is in fact functional to the results presented in 4.1.**

We will replace this title by: "Estimation of the biases of the tracer release method due to the mislocation

of the tracer with theoretical model experiments". We will try to follow the reviewer's suggestion to merge the present 3.6 and 4.1 sections in the new section 3.

**\* Section 4.1**

**See previous comment on section 3.6.**

**\* Section 4.2**

**Page 13: it is redundant to repeat in the text the numbers already reported in the cited Table 2. Please revise this part to avoid such redundancy and repetition.**

We will rework this section to be more concise.

**\* Section 4.3**

**Same as for Section 4.2 about redundancy with Table 2. It would be worth to discuss more in depth why the tracer release method is better than the combined approach for the configuration 1. Because of fewer sources of uncertainties?**

We will also rework this section to be more concise. Yes, with the configuration 1, all the sources of uncertainties for the tracer release method are also sources of uncertainties for the statistical inversion: the measurement error, the difference of time between the C2H2 and CH4 measurements (which implies some uncertainties that were underestimated in the present version of the manuscript), the uncertainty in the background signals (a term which was under-estimated in the present version of the manuscript), and the uncertainty in the debit of methane and C2H2 (which is rather low). Since, the statistical inversion is also impacted by the transport modeling error and the uncertainty in the estimate of the prior and model error statistics, it cannot perform better than the tracer release method in such an experimental configuration. We will better emphasize it.

**\* Section 5.**

**In the results analysis and conclusions there is not a definitive and decisive proof that the combined method provides better and more reliable results than the tracer release method, given the limited number of cases and conditions considered, and the connected uncertainties.**

We do not look for a definitive and decisive proof of the success and superiority of our concept (see our answer to the reviewer's first general comment). We just want to demonstrate, in this study, that it has some potential. For the four tests that have been led, the results, considering the uncertainty estimates or not, were in line with what we expected based on our theoretical concept. For each of the four tests, more than 15 transects of measurements are conducted through the plume. This is similar to past studies on industrial sites (Yoshida et al. 2014). We will better discuss the need for far more tests and studies to evaluate more precisely the robustness and potential of the method and improve it, but, according to us, these results are very promising and quite clear regarding the potential of the approach.

**In addition, the potential ability of the method for multiple sources could not be fully addressed. The authors honestly recognize this and it becomes clear that these are 'preliminary' results and that more experiments are needed.**

Yes, this paper show that in practice, the separation of sources is not straightforward even with the statistical inversion and more studies will be needed to improve it. However, the experiments show that the method can provide more precise estimate than the tracer release technique.

**Thus, the paper is mostly the presentation of a method but not a final test of it, supporting its adoption.**

Yes, we will better insist on the fact that we just propose a new approach and demonstrate its relevance to encourage further studies on it. We should just insist that it also demonstrates some weaknesses of the tracer release approach (and the underestimation of its uncertainties when deriving it from the STD of its estimate transect by transect).

**Figure 2: different colours (or line type) for the curves for different stabilities may better highlight the results.**

We will the color lines in order to better differentiate them.

---

## Author Comment (AC2) · 14 Jun 2017

**In Ars et al., the authors describe a new method for estimating gas emission rates from industrial facilities, by combining 1) tracer flux measurements, 2) Gaussian dispersion modelling and 3) a statistical inversion algorithm. The new method is evaluated using controlled methane/acetylene releases and compared to results from tracer flux. Four tracer placement scenarios are evaluated to demonstrate the improved accuracy of the method in situations where the tracer and the emission source are not perfectly collocated.**

We thank the reviewer for his constructive, detailed and technical comments that will help improve our manuscript.

**GENERAL COMMENTS**

**Overall, I was intrigued by the method presented by the authors, since accurately measuring industrial methane emissions remains a challenge, especially as we attempt to find convergence between bottom-up and top-down measurements.**

We agree that such an estimation remains a challenge. Our objective is to propose a new approach to support it but we do not claim that it has already reached such a maturity after a first study that it should be considered as fully robust. We rather demonstrate its potential and that it can overcome some of the drawbacks of the tracer release technique, which is widely used nowadays for such an activity. Further studies and tests will be needed to improve it and allow for its generalization, including tests with more complex models than Gaussian ones. The introduction and conclusion will better emphasize it.

**However, I am a little skeptical of value of the method in its current form and would like to see more discussion of the method's wider applicability before this paper is published in AMT.**

See above. It is difficult at such a stage to provide an objective evaluation about its level of applicability and our main objective is not to present a specific method with a "rigid" configuration but rather to present a general concept and its potential through a rather simple practical configuration. Section 2 already indicated that a wide range of options could be taken to define the observation and control vectors. The conclusion will now better indicate that the number of options for the transport model is large. This will also open many options for the control of the model to fit the tracer data if using models more complex than Gaussian models.

**Some areas of the method that I think warrant more discussion include the effect of different methane emission rates (only one was tested, 0.4 kg/h)**

Multiplying the number of tests would have provided a more robust evaluation of the very specific configuration of the inversion used in this study. However, again, given our main objective to expose a theoretical concept and demonstrate its potential, and given the difficulty, in practice, to multiply such tests, we find that the number of tests selected was enough. This will be clarified.

Regarding the rate of emissions: see our answers to the reviewer's specific questions about this topic. By using a rather low emission rate, we have challenged the ability to separate the tracer and targeted plumes from the measurement noise and from the acetylene and methane background variations. This impacted both the tracer release technique and the statistical inversions. Due to the linearity of the transport over such distances, results should be better but comparisons between the methods should be similar if using

larger emission rates.

**and the role of meteorology – I am particularly concerned that the authors selectively looked at plumes from very specific atmospheric conditions.**

Yes, the proposed method could have weaknesses in less favorable conditions. The conclusion will better reflect it. The tracer release technique also requires some favorable meteorological conditions, especially a relatively constant wind direction and sufficient wind speed in order to be able to measure an increase of the concentrations through a clear "emission plume". Still, the tracer release technique does not require as homogeneous and stationary local meteorological conditions as our practical implementation of our inversion concept in this study that is based on a Gaussian model. Therefore the result section will now discuss the tracer release method results when using all the transects that can be retained for such a method even though they cannot be retained for the statistical inversion, and show that, in our practical case, these "new" transects do not improve the results from the tracer release technique.

The conclusion will better address this general topic. Of note is that this conclusion section will be expanded into a sort of discussion / conclusion section to better address such points, others asked by the reviewers and to gather some of the discussions that lengthened the previous sections.

**If this new method is being proposed as an "easy-to-implement" method for operators to employ (as it is described in the Introduction), then I would expect such a method to be robust to different atmospheric stabilities. The quality of the writing is excellent, but the authors would do well to streamline the paper so it is less bogged down in text.**

We will improve the introduction and conclusion to ensure that we do not go beyond the objectives and conclusions indicated in our answers above, and that we better present them. We agree that if the method had to stay as "easy-to-implement" as in our practical test with a Gaussian model, it would definitely be challenging to generalize it in terms of the topography, complexity of the facilities and local meteorological conditions of the sites to be studied. More studies and potentially more complexity (using models that are more complex than Gaussian models) will be needed for such a generalization and, if the concept can really lead to operational systems, to find trade-off between accuracy and complexity for being used by operators. However, with this first study we are definitely far from claiming to address such topics.

**I tend to agree with Reviewer #1 who described the writing as "verbose". This complex writing style makes it more challenging to follow the science. Additionally, I think the authors could limit some of the discussion of the methods, particularly the tracer release and Gaussian methods, as these are well-described in the literature, to make room for a more well-rounded discussion of the results, which seemed rushed.**

We will improve the concision of the text especially in sections 2 and gather sections 3 and 4 into a single one and better discuss the results in the light of the four reviews we received for this manuscript.

**Upon making these major revisions, I expect the publication will be suitable for publication in AMT.**

We thank the reviewer for this assessment.

**Specific comments follow.**

**SPECIFIC COMMENTS**

**Section 1 – L105: Is this method easy to implement for operators?**

See our general answer above. This sentence of the introduction was not really aiming at characterizing the specific method presented in this paper but rather at giving a context and long-term objective for the development of its concept and of its practical implementation(s). The method used for our test in this first paper is relatively easy to implement and could definitely be handled by external consultancy groups or by the development department of the operating companies. However, since we acknowledge that first developments may rather go towards complexity than simplification, and since such a consideration is far from the stage of development of our concept, this part of the text will be rewritten.

**How have operators historically monitored their emissions?**

To our knowledge, most of the landfill, waste water treatment plant or gas operators who derived their own methane emission estimates used standard bottom-up product of emission factors times quantity of waste/wastewater/gas processed and/or emission models. In some cases, some operators ordered measurement campaigns by internal (within the R&D department for large operators) or external laboratories with close chambers, tracer release techniques or some completely different measurement concepts (like the recent flux measurement technique based on LIDAR promoted by NPL, Robinson et al. (2011)) to valid or complement such estimates. However, to our knowledge, the use of real measurements is still rather limited.

**If the paper is framed as being in support of industry, then this should be discussed; I am not familiar with many facilities actively conducting tracer flux measurements or those with mobile laboratories to measure downwind emissions.**

See our answer to the previous questions. Industries are definitely interested in such a monitoring of their methane emissions and, we should even mention that, since this study, in the context of projects in collaboration with industrial partners, we have led campaigns and tested our method for their industrial sites (it will be documented in future studies). However, the support to the industry is a bit out of the specific focus of this first paper even though it definitely feeds its context and the long-term goal of the development of our monitoring concept. This first paper is exploratory and conceptual and it is not orientated towards the development of an operational approach. Finally, even though the main field of application should be connected to industrial sites, in principle our concept could be applied to natural "point sources". Therefore, we prefer avoiding spending much time on this topic and, again, the corresponding sentence will be modified.

**Section 2.1 – L175: I am not sure how you have demonstrated that your method provides satisfying results over those distances and methane emission rates compared to what you have tested with your controlled releases – please elaborate.**

We will remove this sentence and add in the conclusion that the method needs to be tested for all of these parameters (distance, spread of the site…) on real sites.

**I am especially interested in how tracer flux and this method differ for large methane emission rates or "superemitters".**

Due to the linearity of the transport, larger sources are easier to invert (i.e. the signal to noise ratio for the concentration measurements is larger) and if the methods succeed in monitoring a given emission rate at a given distance, it should succeed in monitoring a larger emission rate at the same distance (keeping all other emission, topography and meteorological conditions similar). Issues arise for small emission rates, when signals are not high enough to ensure a clear separation between the targeted plume from the noise

of the instrument or from the background variations of the concentrations. This will be discussed in the conclusion.

**Section 2.2 – L180: I suggest that the authors conduct a more thorough literature review, particularly of tracer release measurements conducted in various shale gas basins in the United States. Numerous papers have come out on this subject in the past 3 years.**

We will refer in the section 2 to the studies of Roscioli et al. (2015), Goetz et al. (2015) and Albertson et al. (2016) who used the tracer release method to estimate methane emissions from different types of sites and sought solutions to overcome the non-colocation of the tracer issue.

**Section 2.2 – L210-215: Can the authors speak to how this effect scales with methane emission rate? Does its significance shrink if total methane emissions increase? Or does its importance scale linearly?**

In principle, due to the linearity of transport, it should not impact this specific effect. Larger source just increase the signal to measurement noise (+ variations of the background concentrations) ratio and thus the precision of the results due to the lower impact of measurement noise and uncertainties in the background concentration variations. This will be discussed in the conclusion.

**Section 2.3 L230-235: Can you expand on this more in the text? I find the model justification to be a little lacking.**

We will better insist on the fact that the Gaussian model is a first, simple and low-computational option to test the method feasibility and its ability to improve the estimations when the tracer and the pollutant sources are not well collocated or when there are several sources of the targeted gas. Other options of models are available for more complex topographies but it was not relevant develop a complex inverse modeling scheme with such options for the first tests of our concept in this study.

**Section 2.3 L250-253: Is this detail on urban vs. rural configurations really necessary? Especially if you don't mention what configuration was used in this study.**

This part of the text gives a general presentation of the Gaussian model so we will keep on giving this information but we will try to give it less weight in the corresponding sentence. And we will indicate that we use the rural configuration in section 3.

**Section 2.5 L305-310: I went looking for an explanation of how the spatial offset was treated in Section 3.2, but this section reference Section 2.5. Please make sure this concept is explained. I would strongly caution against routinely referencing other sections, particularly future sections, and instead focus on a linear narrative for the paper.**

Section 2.5 will now be more concise and will not enter into this level of details while this topic will be better addressed in the new section 3.

**Section 3.1 L342-345: If the authors are going to be highly selective of meteorological conditions, then this should be discussed in more detail. What happens on days with low winds?**

When the wind speed is not strong enough, the signal coming from the site is very low and difficult to catch on the roads where the measurements are carried out. Similar wind conditions are requested for the tracer release method. See our answer to the reviewer's general comment on this topic.

**Section 3.2 L360-364: Here are some more cyclical references – I don't think the spatial offset is**

**ever properly described.**

We will gather as much as possible the details and improve the clarity of the text on this topic in the new section 3.

**Section 3.5 L430-435: Choosing stability class based on best fit to the measurements seems suspect to me. How does this choice compare to the estimated stability class using wind speed and insolation metrics? Furthermore, what were the range of atmospheric stabilities during all your tests? Is this method applicable to all stability classes? This is a main point of concern for me and the authors should better justify their decision regarding the Briggs parameterization.**

There could be inconsistencies between the selected stability class and the local meteorological conditions due to the empirical formulation and parameterization of the Gaussian model.

However, for each transect, we have checked that the stability class yielding the best fit between modeled and measured acetylene was systematically in agreement with the measured wind speed according to the Pasquill classification. The solar radiation information may be difficult to have, especially at this scale, and it is thus difficult to choose between the different (in practice for any transect: two) stability classes that were in agreement with the wind speed without the tracer data. We will clarify this and indicate in the text, for each series of measurement, the measured wind speeds and the two stability classes that we could have used according to Pasquill classification and to this measured wind speed, along with the selected class according the model-data tracer comparison. Further tests should be done with stronger wind (> 5 m/s) to make sure the method is applicable to all stability classes as will be discussed in the conclusion.

Finally, even though it did not occur in our experiment, we still think that selecting a stability class that is not consistent with the actual meteorological conditions is not a critical issue as long as the fit (in terms of area and shape) between the modeled and measured tracer plumes is good, since this is definitely a good indicator of the model performance as a function of its parameterization. This will also be discussed in the conclusion section.

**Section 3.5 L472-479: I could not follow this; can you explain how these uncertainties translate into those methane emission rates?**

Some typos made these sentences difficult to read and we apologize for it. The text will be clarified by better indicating the actual values for the sources, the prior knowledge of the inversion on these sources, and the uncertainty in this prior knowledge both in terms of relative uncertainty (compared to the prior estimate) and in terms of absolute value.

**Section 4.2 L584-590: I am not convinced by the argument that the performance of the new model was the worst compared to the actual emission rates due to the low emission source – it seems to me all the other configurations used comparably low emission rates and this problem was not observed. Please provide a better explanation.**

This part of the text discussed the results from the tracer release technique. The errors from this method will not be smaller for the other cases, on the opposite. The measurement errors, the difference of time between the tracer and methane measurements (since the instrument continuously shift between the two types of measurements) and the variations of the acetylene and methane background concentrations are the main source of error for this technique when the methane and tracer sources are perfectly collocated as here (the other source of misfit between the results and expected truth being the uncertainty in the debit of the controled emission which is small). We will better explain that the difference between the estimate and the real flux comes from these factors. In particular, the impact of the difference of time between the

tracer and methane measurements and the variations of the acetylene and methane background concentrations were quite forgotten in the present version of the manuscript while they actually have a critical weight in the results. We will update our computations and results to better account for them, in particular for the configuration 1 where they definitely appeared to have a larger impact, and we will better discuss these sources of errors in the new version of the manuscript.

**Section 4.2 L584-613: This is repetitive of table 2 and does not to be listed off in the text.**

**Section 4.3 L628-652: Again, this is repetitive, I would prefer to see more of an analysis vs. repeating of figures in a Table.**

We will not provide all these numbers in the text and make it more concise. On the other hand, we will expand our analysis of the results as indicated by our answers to the specific comments of the four reviewers.

**Section 4.3 L432: The authors explain the poor performance in configuration 1 does not matter very much due to the fact that the configuration is unrealistic. I am not satisfied with this explanation, if theory dictates that unreasonable or not that configuration 1 should be the ideal case then a good reason should be provided why it was not.**

We will remove this part of the paragraph to avoid misinterpretation and we will rewrite this paragraph to better explain that the tracer release method is supposed to give a better estimate of the emissions than the combined approach in the configuration 1 because in this case the tracer is a better proxy of the transport than the model.

**Section 5: Nowhere in the conclusions (or in the results) do I see any statements on the performance of this method vs tracer flux for a range of methane emissions. This was introduced in the introduction and I do not think it was adequately followed through on. If this method is currently limited to low industrial emission rates it should be expressly stated. As it stands, I think the usefulness of the method is overstated and the authors should be realistic about what their experiments have demonstrated.**

As discussed above, if the method is successful for low emission rates, it should be successful for high ones provided that the size of the source stays the same. We will discuss this in the conclusion. Furthermore, as expressed earlier in our answers to the general question by the reviewer, we do not aim at demonstrating that we have an operational system for monitoring industrial sources yet, and we will better clarify our objective of introducing a new concept and demonstrating its potential through a rather simple practical case (expecting that it could definitely lead to a robust and generalized approach on the long-term).

**Table 1: If I understand correctly, on the days where meteorological conditions were explicitly controlled for, the plume capture rate is roughly 30%. This seems very low to me making me question the robustness of the method. Please comment.**

In order to compare the results of the two methods, we used in this study only transects suitable for both of techniques even if some of them could have been used only for the tracer release method and others only for the combined approach (strictly speaking, there is no "control of the meteorological conditions" in these cases). That explains the low numbers of transect selected. However, we will now discuss the results from the tracer release technique when selecting all suitable transects for this method even if they cannot be used for the statistical inversion, demonstrating that this does not improve the results when using this technique.

Again, in the future, the concept of the statistical inversion should be improved (in particular by testing it with more complex models) and one of the objectives could definitely be to be able to exploit more measurement transects than in this first practical application.

**TECHNICAL CORRECTIONS**

**Section 2.3, L219: Please define acronym "LES" and possibly "CFD", as I am unsure if everyone would know what these are.**

**Section 3.1 L340: Typographical error "serie" Section 3.6: Edit section title to be more succinct.**

These corrections will be done.

---

## Author Comment (AC3) · 14 Jun 2017

L. Golston

lgolston@princeton.edu

Ars et al. introduce a method for inverting small-scale emissions using a statistical framework which incorporates a Gaussian plume model and observations from the tracer method (here with acetylene). This is motivated because in real-world environments the exact location of the methane source could be spread out, inaccessible, or not precisely known, limiting the accuracy of the single tracer release technique by itself. Validation of the combined approach is attempted using a controlled methane and acetylene release experiment in four different configurations.

I read this paper with interest since I am also doing work related to quantifying methane emissions at small scale. From the measurement side, the experiment appears to be well conceived and performed. However, I noticed multiple things that could be clarified or improved on the analysis side. I think there are some key revisions needed to helpful clarify the method and results, along with additional suggestions to improve the overall manuscript that are given in this comment.

We thank M. Goldston for his interest in our paper and his constructive and technical comments. They will help us improve the manuscript, and in particular the result section. Of note is that the conclusion will be expanded into a sort of discussion/conclusion section to gather many of the digressions that lengthened the previous sections and to include discussions asked by the reviewers.

**Major comments**

**1. Physical basis: On the basis for the approach, it makes sense a the tracer releases could, for instance, constrain the dispersion in the Gaussian model when the tracer source is collocated with the methane source. However, when the tracer is positioned farther away, for all the reasons outlined in the paper (lack of a homogeneous / stationary atmosphere), the information from the tracer should become decreasingly useful as one has to rely more heavily on the model results. This is both because the diffusion of air along the acetylene path (obstructions, elevation changes, etc.) may not be the same, but also because a Gaussian model is used to bias correct for differences in downwind distance between methane and tracer.**

The conclusion will better discuss these points. Overall, our aim is to introduce a new concept (calibrating a statistical inverse modeling framework based on tracer release information for the monitoring of emissions from industrial sites) and to illustrate its potential with a simple implementation using a Gaussian model and a practical test for which the conditions are relatively favorable to such a use of a Gaussian model, with relatively stable and homogeneous wind conditions, and in a field with a relatively flat topography and few obstacles (building) that could impact the emission plumes. Considering the scale of this study (with a maximum distance between methane and tracer sources of 60 m), it seemed relevant to make the assumption that the tracer is a good indicator of the atmospheric stability for both the tracer and methane transport. The results confirmed the good behavior of the Gaussian model calibrating based on the tracer data.

The introduction and conclusion will better support future analysis and improvement of such a concept, in particular by using more complex local scale transport models to generalize its applicability (see also our general answers to the anonymous reviewers on these general topics).

For more complex experimental cases, the Gaussian model would definitely miss the impact of the

topography and of variations in space and time of the local meteorology which could make the transport from the tracer source to the measurement location significantly different from the transport from the methane sources to the measurement locations (of note is that we could still run such a model with different wind forcing for different point sources if having relevant measurements, and recombine the full signal for the set of sources thanks to the linearity of the methane atmospheric transport at such a spatial scale). However, in principle, the tracer could be used to control parameters of models that better account for the heterogeneity in space and time of the local meteorology and of the topography during the experiments than the Gaussian model (like CFD/LES ones).

It is difficult (and definitely out of the scope of this first paper on our concept) to assess at which point (e.g. distance between the tracer emission point and the different targeted gas emissions points of the industrial site as a function of the meteorological heterogeneity and of the topography / buildings) it would become difficult to use the tracer to control such models but this issue could be partly handled by emitting tracer at multiple points surrounding the industrial site.

**Yet, the test which performed the worst (in terms of relative difference) for the combined approach was actually Configuration 1, with a collocated tracer, while it performed better for all of the three conditions which should have increased the uncertainty of incorporating the tracer information.**

We can explain it due to the impact of uncertainties associated with the difference in time between the tracer and methane measurements and with the variations of the acetylene and methane background concentrations which have been estimated by calculating the standard deviations of each series without the plume crossings (for example 0.3 and 9 ppb for acetylene and methane respectively for configuration 1). These two sources of errors that we underestimated in the present version of the manuscript and that highly impacted the computations for the first configuration will now be better accounted for and discussed. In particular, the results will be updated due to using a slightly different method for integrating the methane and acetylene plumes and defining the background concentration above which the plumes are computed. It will decrease the impact of these sources of errors in the first configuration

**It makes sense that the uncertainty would be higher for the combined approach than the tracer for Configuration 1, and that the tracer generally was worse for the non-collocated experiments. However, the result for the combined approach is unexpected, both from the perspective of the theoretical basis for the combined approach, and the interpretation of the actual measurement results.**

See our answer above. And, more generally, even though the emission conditions where more favorable in configuration 1 than for the other configurations, the transport and measurement conditions (in addition to the background conditions as point out above) could be potentially more challenging. We can only compare the tracer method and statistical inversion for each configuration rather than results from the different configurations in such a context. It will be better analyzed in the result section and discussed in the conclusion section.

**Does the combined approach account for uncertainty of the tracer technique when used under ideal (collocated) vs. non-ideal conditions?**

Uncertainties for the tracer technique (in our specific case) come from:
1) the problem of colocation between the tracer and the targeted gas
2) measurement errors
3) differences in time between the acetylene and methane measurements
4) variations in the background variations of tracer and targeted gas

Misfits between the estimations and the known emission rate can also be driven by uncertainties in this known emission rate but we did some verifications to ensure that this uncertainty is negligible compared to the misfits obtained in this study.

Uncertainties for the statistical inversion come from: 2), 3), 4) and
5) transport model errors
6) uncertainties in the estimate of the statistics of the errors in the prior estimate of the emissions and of the measurement and model errors.

1) is negligible for configuration 1. In other configurations, strictly speaking, it should impact the tracer technique only (but, actually, it impacts 5) and thus indirectly the statistical inversion; see our answers to the previous comments above). 2), 3) and 4) impact both the tracer technique and the statistical inversion. The estimate of the statistics of measurement and model errors in the statistical inversion system (based on model-data tracer comparisons) should account for 2) and 4) (in addition to 5)). However, in case of 4), it is estimated for the tracer only while the $CH_4$ background variations may be larger than that of $C_2H_2$. At last, 3) is ignored in the estimation of the measurement and model errors in the statistical inversions.

We will clarify these points in the new version of the manuscript.

**Why where the combined results better (again, in terms of relative and absolute difference) when the tracer is used under less ideal conditions? More physical insight into how the combined approach works would be very helpful.**

See our answers above. The new results will show a different picture since better addressing the sources of errors associated with the background concentrations and alternative measurement of $C_2H_2$ and $CH_4$.

**2. Organization: In general, there is unusually frequent referencing back and forth between Sections 2 and 3 and at one point even a "circular link" between section 2.5 and section 3.2 about the time lag, with neither quite containing the indicated information. Section 2 largely reviews the literature on these three techniques separate from the details of this paper, and could easily be condensed or even combined with section 3 since there are a lot of similarities between the two, and I think it would make it easier for the reader to understand specific aspects of the way the approach and experiment were conducted which is currently tedious going back and forth between the different sections and subsections.**

We will improve the concision of the text especially in section 2, avoid redundancies and gather sections 3 and 4 into a single one, within which we should merge some of the subsections that were previously split between section 3 and 4. Section 2.5 will now be more concise and will not enter into its present level of details while the corresponding topic will be better addressed in the new section 3.

We want to keep the general structure as it is and gather sections 3 and 4 into a single section 3 since we feel that this paper has definitely two components: the definition of a new theoretical framework in one hand (the present section 2), and its evaluation through an academic experiment (the present sections 3 and 4). Many discussions and details will be aggregated into the last section which will now be a discussion / conclusion section rather than a short conclusion section.

**3. Relationship to other literature: The effect of non-collocation, including distance of the measurement and magnitude of non-collocation, and the effect of being confined to the road which prevents non-orthogonal slices were discussed. These are all important issues for subsequent people using this method or similar experiments, and is also related to one of the conclusions [L676 - L679],**

**so several recent papers would also be valuable to cite on these topics:**

**Goetz et al. 2015, Environ. Sci. Technol. (doi:10.1021/acs.est.5b00452) investigate issue where tracer not collocated and employs a correction based on the Gaussian plume**

**Roscioli et al. 2015, Atmos. Meas. Tech. (doi:10.5194/amt-8-2017-2015) also look at effect of the tracer and source not being collocated using the dual tracer framework to bracket possible errors, which is an alternative approach to what is given here and is likely applicable to similar types of sites**

**Albertson et al. 2016, Environ. Sci. Technol. (doi:10.1021/acs.est.5b05059) employs Bayesian framework and also specifically discusses and gives a correction for the issue of the road not being orthogonal to the wind direction also based on a Gaussian plume formulation**

We will cite these relevant papers when discussing the imperfect collocation of the tracer and methane sources and its effect on the flux estimations with the tracer release method in the section 2.2 and 2.5.

**4. Details missing that are important for understanding the approach/experiment:**

**- It is vague what information from the tracer is combined with the Gaussian, is it just the (rather coarse) adjustment of the stability class A-F, or more fine scale impact on the parameters ($\sigma y$, $\sigma z$, and/or wind)? It would be helpful to see how the parameters were actually affected during these experiments**

The wind forcing is imposed based on our local wind measurements. The tracer observations are used for the selection of the model stability class. It appeared that the optimal stability classes in terms of fit between the modeled and measured tracer were systematically in agreement with the range of stability class corresponding to the measured wind speed (according to the Pasquill stability classification, see 2.3). The horizontal and vertical Gaussian plume standard deviations $\sigma y$ and $\sigma z$ are then set-up according to the class of stability. Even if it seems coarse, the model showed a good ability to reproduce the measured tracer concentrations with the selected stability classes while being highly sensitive to the choice of these stability classes.

We will improve the clarity of the text on this topic.

In the future, we expect to use tracer data to control for more complex parameters in potentially more complex models (directly controlling the $\sigma y$ and $\sigma z$ of the Gaussian plume spread might be a good option). However, again, in this first paper, we wanted to demonstrate the potential of our approach with a rather simple implementation case before increasing the level of complexity.

**- How the prior uncertainty is determined is not discussed, and the basis for model + observation uncertainty only briefly**

The configuration of the prior uncertainty is set high to reflect a lack of prior knowledge on the source as when investigating actual industrial sites. We will better describe and justify it. The presentation of the prior values also needed improvement and clarification. More critically, we will better explain in the conclusion section that the weight of the prior information is relatively low in our statistical inversions (thanks to the choice of high prior uncertainties).

The model and measurement uncertainties are directly derived from the model-data tracer comparison. It is a critical part of our method that was actually discussed in several paragraphs so we will better explain and highlight it.

**- Both the method and results for the "multiple sources" inversion is brief other than that the plume is divided into five slices. Can a figure be added to illustrate how this works? Does using five slices mean up to five sources can be quantified? Can the approach resolve multiple sources when the plumes are overlapping, or only when they are basically non-overlapping?**

An example of the division of the emission plume and of the integration of the area under each slice was given in the figure 6 (bottom figures).

Since we use a statistical inversion, we do not have to follow a strict relationship between the number of sources than can be solved for and the number of slices (we can solve for more or less sources than the number of observations i.e. slices assimilated by the inversion system). Still, if willing to bring a strong constraint on individual sources, one must have a sufficient number of slices bringing independent piece of information. Having a large number of slices ensures having maximized the potential for catching the different piece of independent information in the signal. But it can give a critical weight for measurement errors, turbulent patterns or background variations, which would dominate in some of the observations. Selecting 5 slices here was considered as a safe trade-off between these two aspects.

If the plumes of all sources strongly overlap, there would hardly be any source of information for individual sources in the signal, and these sources would be difficult to quantify separately. Increasing the number of slices would not help solve for this problem. The level at which the slices can provide independent information for individual sources is difficult to anticipate but it is characterized in the posterior uncertainty that is diagnosed by the inversion system. The more the plumes overlap and the highest the posterior uncertainties on these estimates will be, and the more negative the posterior correlations between these posterior uncertainties in individual sources will be. The analysis of these posterior uncertainties will better highlight it and this general topic will be better discussed in the manuscript.

In a more general way, the analysis of the results will be expanded following the different comments by the four reviewers.

**5. Table 2: Two things stand out about Table 2, where results are given from the controlled release experiment.**

**First, why is there no row given for a Gaussian inversion, separate from combined approach? This is key information in evaluating the difference between the tracer release, Gaussian, and combined approaches.**

We could have considered deterministic inversions using the Gaussian model. However, we are not sure that it brings a lot of insights: if using the Gaussian model with a configuration ignoring the tracer results, one would get relatively poor estimates of the emissions. If using a configuration minimizing the misfits between modeled and measured tracer, one would already have used one of the components of our proposed concept (there would already be some combination of approaches). And, when targeting two methane sources, one would have to separate the measured signal into two slices to ensure that the transport is invertible which would be both inspired but different from the choice made in the statistical inversion.

In a general way, we do not think that deterministic inversions based on model handle all of the issues connected to the tracer release technique that are discussed in section 2, so we prefer to compare the tracer release technique to the statistical inversion only. Furthermore we do not really see how analyzing the behavior of such a technique would bring insight on that of the statistical inversion (which would use a different control or observation vector). Therefore we prefer not applying and discussing results from a

deterministic inverse modeling framework. We will briefly mention it in the conclusion.

**Secondly, the uncertainty given for the combined approach is extremely small, several times smaller even than the controlled release uncertainty for Configurations 1, 2, and 3 which does not make sense.**

We will better:

- remind that it is difficult to compare the uncertainty estimates for the tracer release technique and that of the statistical inversion (since they rely on different assumptions even when they are assumed to encompass similar sources of errors)
- discuss the fact that the posterior uncertainties from the statistical inversion are likely under-estimated; our revision of the results (see above) will not change this general fact. Among other explanations, it can be due to an under-estimation of the model and measurement errors due to ignoring spatio-temporal correlations of this error for individual slices, due to ignoring the difference in time between tracer and methane measurements, or due to underestimating the impact of variations in the background concentrations when looking at tracer concentrations (while the variations can be larger for methane). More generally, the translation of model errors when simulating the tracer plume into estimates of the statistics of model error when simulating the methane plume(s) relies on assumptions that may be too optimistic in our experimental case.

**This is also noticeable in that for Configuration 2, the statistical chance of the uncertainty ranges 428 ± 7 and 464 ± 1 overlapping is 1-in-a-million, and for Configuration 1 the change of 382 ± 7 and 472 ± 2 overlapping essentially impossible. It is said all of the sources of error are considered, but this is clearly not the case - more thought should be given into how to derive a more representative uncertainty range for the combined approach.**

See our answer above. All of this will be better analyzed. A very specific source of errors was actually ignored (the difference in time between tracer and methane measurements). Furthermore, even if a source of error is taken into account theoretically, in practice, it can be very difficult to quantify its impact.

**6. Clarification on use of 'transient': The time dependence and transience of the problem is mentioned several times, but I do not think a clear explanation is given.**

When using this term, we were just referring to the fact emissions for the industrial sites can vary in time over annual / seasonal timescales to daily (and even hourly and smaller) timescales. By conducting relatively short term campaigns (typically on specific days and even over time windows of less than few hours) one may only provide a picture of the emissions that cannot easily be extrapolated into an annual rate for the industrial site, while inventories usually report and focus on such annual rates. This will be clarified.

**- In Figure 6 was the release not being run continuously? How was the time dependence of the prior derived? Additionally, the title says "Gaussian plume", which is formulated for a continuous, averaged value, not a transient release. Clarification is needed on this issue.**

Yes, the tracer and methane release are constant in time and if targeted an unknown source, we would use a tracer release that is constant in time. In cases for which the targeted source would vary in time during the measurement time window, this would challenge the tracer release technique. It would challenge our practical implementation of the statistical inversion but we could imagine a more complex statistical inverse modeling set-up (with models that can clearly account for the temporal variation of the sources) to account for potential variations in time of the targeted source. This will be better discussed in the last

section.

**Minor comments L487-491 and L556-559: Both good points L492-495: Sentence could be clarified**

We will clarify it by merging 3.6 and 4.1 in the new section 3.

**Table 2: including the bias due to mislocation as part of the " +/- " does not make sense since it is not a random error.**

We agree with the reviewer. We will change our way to present these uncertainties as follows: ± standard deviation of the random uncertainty derived from the variability of the results from one transect to the other one (bias due to the mislocation of the tracer ; estimate of the total uncertainty based on the RSS of these two terms)

**Also what about the uncertainty of the bias, since presumably this is non-trivial?**

All estimates of uncertainties and biases are uncertain (see above). However, our protocol to derive estimates of uncertainty and biases are deterministic and detailed explicitly. The numbers given correspond to these protocols and from that point of view they do not require any uncertainty bars.

**Table 1: one of the columns says 'wind direction (degree)', when that column is given in letters rather than degrees**

We will remove (degree) from the table.

---

## Author Comment (AC4) · 14 Jun 2017

**General Comments:**

**Overall, we think this is a rigorous methods paper and does a good job of describing a novel combination of tracer and dispersion methodologies. The descriptions of the experiments and methods are clear, and will be well understood by the community, both on the experimental side and on the dispersion side. We think this paper should be published after addressing the comments below.**

We thank S. C. Herndon, J. R. Roscioli and T. I. Yacovitch for this general assessment of our study and manuscript and for their constructive and technical comments that will help improve both our analysis and the text of our manuscript.

**1. Please cite some important related research that explores tracer mislocation effects:**

**Goetz et al. used a similar combined methodology in 2015 for tracer release experiments at wellpads. Goetz, J. D.; Floerchinger, C.; Fortner, E. C.; Wormhoudt, J.; Mas- soli, P.; Knighton, W. B.; Herndon, S. C.; Kolb, C. E.; Knipping, E.; Shaw, S. L.; et al. Atmospheric Emission Characterization of Marcellus Shale Natural Gas Development Sites. Environ. Sci. Technol. 2015, 49, (11), 7012; DOI: 10.1021/acs.est.5b00452.**

This article will be cited in the section 2.5 to mention the possibility of using a Gaussian model to determinate a correction factor in order to take into account the mislocation of the tracer in the tracer released method.

**Roscioli et al. Performed an extensive error analysis of the impact of tracer mislocation using dual-tracer release methodology. Additional methods of calculating the methane/tracer ratio are also described. Roscioli, J. R.; Yacovitch, T. I.; Floerchinger, C.; Mitchell, A. L.; Tkacik, D. S.; Subramanian, R.; Martinez, D. M.; Vaughn, T. L.; Williams, L.; Zimmerle, D.; et al. Measurements of methane emissions from natural gas gathering facilities and processing plants: measurement methods. Atmos. Meas. Tech. 2015, 8, (5), 2017; DOI: 10.5194/amt-8-2017-2015.**

We will cite this article in section 2.2 to present the different methods of calculating the methane/tracer ratio and the impact of the tracer mislocation in the dual-tracer estimates. This article and the one suggested above will also likely be cited in the new conclusion too since the latter will be expanded into a sort of discussion / conclusion section (in order to include some of the discussions that previously lengthened sections 2-4 of the present version of the manuscript, and in order to lead the discussions asked for by the four reviewers).

**2. Please address some potential problems with the experimental measurements: Have experimentally measured mixing ratios been calibrated? Please briefly mention calibration procedure.**

Before the experiment, the instrument has been tested in the laboratory. It showed a good linearity over a large range of mixing ratios and a good stability over time with small dependency to pressure and temperature. The feedback from other instruments of the same type was that after a shutdown an offset could appear. To control for that offset, we measured a  gas with a known mixing ratio (calibrated with a multi-point calibration in the laboratory) before each series of measurements in order to ensure the good analytical performances of our instrument. No offset was seen after shutdowns in our case. Moreover, in the tracer released method and the combined approach presented in this study, we are interested in the increase of concentrations (the amplitude of the plumes) due to the tracer and targeted point sources above the background signal more than in the absolute value of the measurements. Thus, an offset of the

measured concentrations will not impact our estimates. This will be clarified in the new manuscript.

**Line 370-375: The flow meter you refer to is an analog measurement of flow rate, so we don't think you have a time-resolved record of the flow rates. While you have a final check of the release by mass difference, do you have any estimate of the variability of the flow rate over the course of the experimental measurements? This could be substantial, particularly with acetylene releases, which can vary as the cylinder cools. In such a case, the flow rate may appear the same on the flow meter, but the actual mass flow may be different.**

We do not have a time-resolved record of the flow rates during the experiments. However each cylinder has been weighted before and after each series of measures and the flow rate calculated with the mass difference was systematically in good agreement with the flow rate read on the flow meter. Therefore we have no reason to believe that there was an important variability of the acetylene and methane release during our experiments.  This will be discussed in the new manuscript.

**Paragraph starting at 441: When multiple instruments are sampling in a mobile vehicle, they will often have different inlet times (lag due to air intake).**

In our case, we use one instrument only and we assume that the inlet time for acetylene is the same as for methane. However, we will now better discuss the fact that, since our instrument measures alternatively methane and acetylene, the measurement times for the two species are slightly different which is a source of uncertainty for both the tracer method and statistical inversions.

**This can be corrected based on experimental measurements, for example by delivering an excess of nitrogen or zero-air to the inlet tip, and monitoring the instrument responses. Has this been done experimentally? How does the time-shifting of data based on simulated plumes compare to the experimental lags?**

The inlet time of our system has been tested during previous experiments by delivering an excess of carbon dioxide to the inlet and measuring the response time. The estimated inlet time is about 10 seconds. However, as will be better clarified in the new manuscript, the difference between the exact effective wind corresponding to the actual plumes and the measured wind used to force the Gaussian model, and modeled concentrations with inlet time correction does not always perfectly fit with measured concentrations due to slight changes in the wind conditions. That is the reason why we rather use the experimental lag to correct the model-data comparisons for each transect. This will be better discussed in the new manuscript.

**3. Why are ideal tracer ratio experiments not producing acceptable results?**

**Config. 1 tracer release method estimates before Gaussian correction (e.g. Table 2, middle row) overestimate the true release rate by 19%, when it should produce accurate results without the need for correction (see discussion below for more detail). This discrepancy casts doubt on the biases of the remaining 3 configuration tests. The calibration of mixing ratios and the consideration of errors in the instantaneous flow rates (see above) should be considered.**

Our explanations is that it is connected to two sources of errors that likely impacted the computations on the first measurements series for configuration 1 than the following ones, and that we under-estimated in the present version of the manuscript:

- the slight differences in time between the C2H2 and CH4 measurements by the same instruments (see above)

- the variations of the CH4, and to a lesser extent C2H2, background concentrations estimated by calculating the standard deviation of both series without the plume corssings (for example 0.3 and 9 ppb for acetylene and methane respectively for configuration 1).

Both raise uncertainties in the computation of the plumes above the background and in the comparison between C2H2 and CH4 concentrations.

These sources of uncertainties will now be better accounted for and discussed. We will update the results to decrease their impact by interpolating the CH4 measurements to the C2H2 measurement time before comparing the plumes, and by selecting another method for defining the background concentrations (by computing their average in parts likely not impacted by the plumes rather than using the 5$^{th}$ percentile) which, in principle, should slightly decrease the sensitivity to their variations. Here is the new table 2 corresponding to this update of the results:

Table 1 – Methane emission rates of the different controlled release configurations estimated with the different approaches and methane fluxes actually emitted during these tests. The uncertainties given with the tracer release method are detailed as follows: total uncertainty (standard deviation of the random uncertainty derived from the variability of the results from one transect to the other one ; bias due to the mislocation of the tracer).

|  | Configuration 1 | Configuration 2 | Configuration 3 | Configuration 4 |
|---|---|---|---|---|
| Controlled methane release (g.h$^{-1}$) | 382 ± 7 | 428 ± 7 | 360 ± 7 | 482 ± 7 |
| Tracer release method estimates (g.h$^{-1}$) | 434 ± 23 (0 ; 23) | 564 ± 120 (296 ; 416) | 321 ± 51 (131 ; 182) | 804 ± 160 (352 ; 512) |
| Relative difference to the control release (%) | 14 | 32 | 11 | 67 |
| Combined approach estimates (g.h$^{-1}$) | 441 ± 6 | 358 ± 2 | 386 ± 2 | 462 ± 34 |
| Relative difference to the control release (%) | 15 | 16 | 7 | 4 |

**Lines 584 – 590: Something seems wrong with this plume analysis. You are well above the detection limit (at least looking at Figures 4 and 5), so the explanation of why the tracer analysis is not accurate does not make sense.**

As mentioned above, we will develop the effect of acetylene and methane background variability and of the slight differences in time between the C2H2 and CH4 measurements by the same instruments to explain the difference between our estimate and the actual emission rate instead of the low emission rates.

**Configuration 1 should be the ideal case. Please rule out instrument calibration errors (both in measurement and release equipment), unit errors (temperature and pressure of measurement), and other experimental issues.**

We will remove the sentence "this misfits associated...." (line 588-590) and explain that the difference between the difference between our estimate and the actual emission rate comes from measurement errors, the difference in time between acetylene and methane measurements, but also from the variations of the background CH4 and C2H2 signal, which cannot be accounted for correctly based on a single "background" value, as in this study. As mentioned previously, such a background computation and the corresponding results will be updated in the new version of the paper to decrease the impact of these variations, and this topic will be discussed.

The accuracy of the sensor used for temperature and pressure measurements is ±0.3 °C at 20 °C and ±1 hPa.

**Dual-tracer release has been used in the past to quantify bias. When they are collocated, the emission rate of one tracer as derived from its downwind ratio to the other is found to agree extremely well with the known mass flow (much better than the 19% error that is observed here). See Figure S4-4 of the supporting information for Allen, et al, Proc. Nat. Acad. Sci. (PNAS), vol 110, page 17768 for an example. It is also shown in the attached figure. There, the downwind ratio of two collocated tracers (C2H2 and N2O) is within 1% of the ratio of their mass flow rates (Figure S4-4c). If N2O were replaced by CH4, the scenario in Figure S4-4 would be experimentally identical to configuration 1 of this manuscript. The tracer release-derived CH4 flow rate would then be within 1% of the known flow rate. This level of agreement for collocated sources is routine in the field. Therefore, the 19% deviation observed here strongly suggests that there is an issue with the measurement method (instrumental), or tracer/methane flow rate.**

See our answers above.

**If configuration 1 were viewed a "control", then the +19% bias indicates that other aspects of the measurements are +19% biased, not the tracer method inherently. In that case, it should be used as an bias offset – that is, the +29% bias for configuration 2 should actually be 29%-19% = 10%, for configuration 3 should be 17%-19% = -2%, and for configuration 4 should be 58%-19% = 39%.**

As indicated by our answers above, we do not believe that the 19% reflect a systematic misfit between our known and targeted rate of the CH4 emission and the perfect estimate that we could expect based on our measurement protocol. We rather see it as a random uncertainty associated with random sources of errors in the background concentrations and in the measurements, which should be associated with our estimates rather than corrected for in such estimates (using our knowledge of the true CH4 emissions), and whose impact will not be the same between the different configurations. So we prefer not following such a suggestion (of note is that the discussion on whether error from configuration 1 should be reported to other configurations in term of relative vs. absolute numbers could be very difficult).

**Lines 605 – 613: Please better explain why this tracer ratio method is not working, given that earlier you say that position errors perpendicular to the wind should not have such a large effect. Was the wind varying?**

During this series of measurements, the wind came from the north-east (there is a mistake in table 1 that will be corrected) and these conditions the shift between the sources is not only lateral. That is the reason why the error due to the mislocation of the tracer is important. We will now better discuss the fact that lateral shifts of the sources perpendicular to the wind direction, in theory, should not impact the results of the tracer release method, even when having several targeted sources, due to the linearity of the estimation, but that configuration 4 did not correspond to such a situation.

**Also, the "configuration 4" panel in Figure 5 shows only one methane plume. Given that there are two CH4 sources, why don't you see two CH4 plumes, or at least a broad CH4 plume? We would expect the two CH4 plumes to look like a composite of the C2H2 and CH4 plumes in configuration 3, which has identical separation.**

Actually, in Figure 5, as for configuration 1, we did not select a representative figure for configuration 4. It corresponds to a case for which the 2 methane plumes seem to perfectly overlap despite the shift between the two methane sources (we can still notice that the methane plume is wider than the C2H2 one). We will replace this figure by another one showing a clearer separation of the methane plumes (even though they still partially overlap) and the acetylene plume collocated with one of the methane plumes.

**4. Initial guesses for optimization**

**Paragraph at 276 & Line 474: Please explain how initial guesses completely independent of measurements can be done in practice, particularly for emission sources where the expected emission magnitudes may span many orders of magnitude (e.g. factor of 100, instead of 80%). This would be the case, for example, for certain oil and gas emission experiments. Is it possible to use tracer release result (without any dispersion corrections) as the prior?**

Such an initial guess for a given industrial site could be given by the product of typical emission factors times for the sector of activity times the level of activity of the industrial site. But yes, this knowledge could be very poor for the examples provided by the reviewers. This is why we use a very high (80%) prior uncertainty in our inversion system. Such an uncertainty, in practice (in our experiments), gives a small weight to the prior estimate in the inversion process, and the inversion results are actually very weakly sensitive to the choice of this prior estimate. It will be discussed in the new manuscript.

The statistical inversion framework assumes that the uncertainties in the prior estimate, in the model and in the observations are fully independent, which would not be the case if the prior estimate is based on the tracer and methane measurements which respectively feed the model and are used as observations for the statistical inversions. We would definitely account twice for such data if using the results from the tracer method as a prior estimate of the inversion.

**Specific Comments:**

**Lines 191 - 196: Other methods of calculating the ratio are commonly used, notably taking the slope of the plot of the methane vs tracer plume signals. This is generally found to be more precise than measuring the area under each plume, because it does not depend upon choice of background. See Roscioli et al: Roscioli, J. R.; Yacovitch, T. I.; Floerchinger, C.; Mitchell, A. L.; Tkacik, D. S.; Subramanian, R.; Martinez, D. M.; Vaughn, T. L.; Williams, L.; Zimmerle, D.; et al. Measurements of methane emissions from natural gas gathering facilities and processing plants: measurement methods. Atmos. Meas. Tech. 2015, 8, (5), 2017; DOI: 10.5194/amt-8-2017-2015.**

We will mention this option and cite this paper when discussing the options for the observation vector.

**Lines 225 : There is considerable debate in the atmospheric modeling community on the timescale appropriate for the canonical A-D stability classes (e.g. Pasquill- Gifford stability classes). The consensus seems to be more on the order of 10-15 minutes than 1-2 minutes. See, for example: Fritz, B. K.; Shaw, B. W.; Parnell, C. B. J. Influence of meteorological time frame and variation on horizontal dispersion coefficients in Gaussian dispersion modeling. Trans. ASABE. 2005, 48, (3), 1185; DOI: 10.13031/2013.18501**

We agree that the selection of the stability class per transect as a function of the continuous wind measurements based on such a table is not perfect, which is why we would have been ready to use a class of stability that does not fit with the wind speed (according to the Pasquill stability classification) in order to get the best fit with C2H2 data as possible. However, it appeared that the stability classes that provided the best fit with C2H2 data were systematically part of those corresponding to the measured wind speed. This will be better discussed.

**Line 302: ". . . spatial offset between the measured plume and the actual plumes due to the lag between the air intake and the concentration measurement." I recommend adding the parenthetical statement: (also known as inlet lag or inlet time)**

We will use this term in the paper every time the topic will be discussed.

**Line 340: correct "serie" to "series"**

This will be corrected.

**Line 351: While the instrument reporting time is noted (2 seconds), the instrument response time is not. Furthermore, the data depicted for configuration 1 in Figure 5 suggests the time response for the two channels is not fully matched. Is the instrument reporting all mixing ratios simultaneously, or does the instrument sub-divide the 2 second interval to quantify each of the noted species.**

As mentioned before, we will be better discussed the inlet time of our system in the new manuscript. The instrument does not measure both species at the exact same time and there is actually more measurements of acetylene than methane (methane is measured approximately every 2 seconds and acetylene every seconds). This will be better described in the analytical equipment part. In particular, as discussed above, we think that this interval between the acetylene and methane measurements is a significant source of uncertainty in our calculations.

**Line 429: Is the 5th percentile of transect concentrations sufficient to determine baseline? Do the results change if the 2nd percentile, or 10th percentile is used?**

The definition of the background (or baseline) will indeed impact our estimates and we assume that it is one significant source of uncertainty in our experiments (see our answer to comments on configuration 1 above). We now think that taking a specific percentile (i.e. a given data) was not the best option for this. This is why we have updated our results with a new background estimated as the average of the concentrations measured before and after the plumes and the new manuscript will discuss this topic of the sensitivity to the definition of the background.

**Lines 561 to 567 and Figure 3: Why not show these results as a function of relative distance downwind, i.e. (distance between source and tracer)/(distance between site and measurement)? The distances in meters shown cannot be easily generalized. I further suggest putting panels a) and b) on the same vertical scale, and adding gridlines at the same intervals for both.**

We agree with this suggestion but we will still provide both relative and absolute distances since the actual atmospheric transport processes will change depending on the absolute distance so that we should be very cautious if generalizing such results. We rather take it as an illustration of the amplitude that the bias can reach in a rather simple experimental case.

**All Figures: Please increase font size of all labels and numbers so that they are readable at the width of a normal sheet of paper.**

OK

**Figure 4: Consider showing only Figure 5 (representative plume transects) instead of the whole data set. These full data results, depicted in Figure 4, might be better left to the supplementary info. Even better, consider publishing these results as a test dataset. If Figure 4 is to remain, then the vertical axes should be rescaled to see all of the plume intensities.**

We would like to keep Figure 4 to illustrate the variability of the background, especially for the methane. We agree about removing the highest measured concentrations that do not correspond to actual plume crossings, but to measurements close to the cylinders when we checked the stability of our emissions. These concentrations were so high that the actual plume crossings could not be seen properly.

**Figure 6: Figure 6 is the most important figure of the manuscript and needs to be reformatted for**

**legibility. It is currently much too small. Consider perhaps a small cartoon drawing of "config. 1" and "config 4". To clarify the difference between the 2 sets of results shown. Please also label the meaning of the shaded area ("concentrations" plots) in the legend or figure title.**

We will improve this important graph by increasing labels size, better indicate the difference between the top and the bottom graphs and explain that the shaded areas correspond to the uncertainties of the modeled concentrations.

**Line 565: Instruments with lower levels of detection than the one used here are available. Please alter this statement to reflect this fact: e.g.: ". . .signal to measurement noise ratio would likely be too small, using these instruments, to derive. . ."**

We will follow this suggestion.

**Section 4.3, Lines 614: Please reference Goetz et al, who in 2015 used a similar approach to this.**

As mentioned previously, we will cite Goetz et al. in the section 2.2 and 2.5.

**Line 651: It is possible, (see Roscioli et al.) in some cases, to co-locate tracers and emission vectors for real experiments. Please rephrase "which can hardly occur"**

Fugitive emissions (like leakages) that are transitory, or affecting poorly reachable areas or complex buildings, unexpected sources, but also widespread and heterogeneous sources (e.g. livestock in the building of a farm, basins in waste water treatment plants or cells in landfills for which emissions are not homogeneously distributed) makes it difficult to know perfectly the distribution of the emissions within a site. This sentence will be modified and extended to clarify it. In particular we will replace "which can hardly occur" by "which is not always easy in real cases".

---

## Referee Report (RR1)

In Ars et al., the authors describe a new method for estimating gas emission rates from industrial facilities, by combining 1) tracer flux measurements, 2) Gaussian dispersion modelling and 3) a statistical inversion algorithm. The new method is evaluated using controlled methane/acetylene releases and compared to results from tracer flux. Four tracer placement scenarios are evaluated to demonstrate the improved accuracy of the method in situations where the tracer and the emission source are not perfectly collocated.

**Resubmission review comments**

Upon re-reviewing the paper, I am satisfied with the edits and responses to review comments. Given current uncertainties that were pointed out in review and were not addressed with additional experiments due to other limitations, I agree with changing the description from "method" to "concept". Provided the editors of AMT are amenable to publishing "concepts/frameworks", then this manuscript should be published subject to minor technical corrections.

All the suggested technical corrections are towards making clearer figures. The editing and revisions to the main text were excellent and I could not find any errors.

All figures: perhaps adding a descriptive word would help keep track of the different configurations. For example, configuration 1 could be relabelled 1: co-located tracer, 2: upwind tracer, 3: lateral tracer, 4: multiple sources.

All figures: check figure resolution, many appeared blurry to me

Figure 1: axis tick frequency in the methane plots is inconsistent between the top panel (configs 1, 2) and bottom panel

Figure 2: I found this difficult to interpret without a careful look. I would zoom in more on the tracer/methane locations. It also difficult to see black text on top of the map

Figure 3: This figure is in French, and should probably be translated to English.

Figure 5: axis labels too small, and blurry

---

## Author Response (AR2)

**In Ars et al., the authors describe a new method for estimating gas emission rates from industrial facilities, by combining 1) tracer flux measurements, 2) Gaussian dispersion modelling and 3) a statistical inversion algorithm. The new method is evaluated using controlled methane/acetylene releases and compared to results from tracer flux. Four tracer placement scenarios are evaluated to demonstrate the improved accuracy of the method in situations where the tracer and the emission source are not perfectly collocated.**

**Resubmission review comments**

**Upon re-reviewing the paper, I am satisfied with the edits and responses to review comments. Given current uncertainties that were pointed out in review and were not addressed with additional experiments due to other limitations, I agree with changing the description from "method" to "concept". Provided the editors of AMT are amenable to publishing "concepts/frameworks", then this manuscript should be published subject to minor technical corrections.**

We thank the reviewer for taking the time to review the new version of our manuscript and for this positive assessment. We remind that our paper provides a practical implementation for our concept and first tests with real measurements, even if in favorable experimental conditions, so that it cannot be summarized as a concept paper.

**All the suggested technical corrections are towards making clearer figures. The editing and revisions to the main text were excellent and I could not find any errors.**

We will take into account these comments to improve the figures of our manuscript.

**All figures: perhaps adding a descriptive word would help keep track of the different configurations. For example, configuration 1 could be relabelled 1: co-located tracer, 2: upwind tracer, 3: lateral tracer, 4: multiple sources.**

We added the description of the different configurations on the figure 1,2 and 5 but also on the table 2 to help the reader remember these configurations.

**All figures: check figure resolution, many appeared blurry to me**

The resolution of the figures has been improved (especially those with subfigures).

**Figure 1: axis tick frequency in the methane plots is inconsistent between the top panel (configs 1, 2) and bottom panel**

We changed the axis tick frequency and used the same one for of each plot. We adapted the color of the points indicating the position of the sources to this new background.

**Figure 2: I found this difficult to interpret without a careful look. I would zoom in more on the tracer/methane locations. It also difficult to see black text on top of the map**

We zoomed into the experiment area and increased the transparency of the map to improve the readability of the figure.

**Figure 3: This figure is in French, and should probably be translated to English.**

This figure has been replaced by its English version.

**Figure 5: axis labels too small, and blurry**

We increased as much as possible the size of the axis labels. We also applied similar corrections to Figure 6.

**Anonymous Referee #2**

**Suggestions for revision or reasons for rejection**

**The manuscript is indeed improved and the authors made a thorough effort to address properly the several comments received and the discussion proposed during the first review process. I appreciate that the authors had particular care in responding to the scientific issues raised by the Reviewers and Colleagues.**

**Now the presentation of the topic and methods are in general better explained and organized, and the discussion of the findings and results is more detailed.**

We thank the reviewer for taking the time to review the new version of our manuscript and for this positive assessment.

**I find still too dense the manuscript itself. It is not a matter of number of lines, but of the risk of missing the key points and findings. More focused and concise statements, in particular when arriving at the discussion and conclusions, would help highlighting what are the real and final outputs of this study. It is difficult to give suggestions about this aspect, since the way of telling is a personal attitude, but I think it is worth to submit this issue to the attention of the authors.**

We rewrote and shortened the long sentences of the previous manuscript to make them more understandable and concise for the reader. We also added and modified some sentences in the abstract and in section 4 to better highlight the key results from this study.

**A few more additional comments are reported hereafter.**

**1. There are some long sentences that make the reading difficult, for instance: lines 76-80, 90-94, 146-150, 235-240 etc. Please consider to rephrase them: in general, all sentences longer than, let's say, three lines might be probably better streamlined.**
We have rephrased the longest sentences to help the reader to follow the argument.

**2. Lines 72-74: this sentence is not much clear, at least to me; I could not get what in practice is done.**

The sentence has been changed into "Moreover, when targeting several sources, this technique relies on the mathematical inversion of a square matrix characterizing the atmospheric transport that links the set of sources to the observation data. This artificially requires extending or limiting the number of observation data from the measurement series to the number of sources to be quantified.".

**3. Line 192: should 'h' (observation operator) be a capital letter?**

In the case of the tracer release method and when targeting one emission rate only, the observation operator used is a scalar. We think that 'h' is more appropriate to refer to such a scalar and to better differentiate against situations when the observation operator is a matrix **H**, even though, strictly speaking, the H (not bold) notation was also eligible.

**4. Line 700: 'can ranged' should be 'can range'**

Done

**5. Line 828: 'in any cases' should be 'in any case'**

Done

**6. Figures 2 and 4 are not cited neither commented in the text.**

Figure 2 is cited and commented line 524 and figure 4 line 666.

---

## Author Response (AR3)

**Dear authors,**

**Thank you very much for the revisions. The reviewers had suggested only minor changes and I consider your responses and amendments as adequate.**

**However, before being able to accept your publication, I have quite a large number of additional suggestions/corrections, most of them addressing issues with grammar and language, but also a few comments/questions on the content. I don't expect you to respond to all these changes point by point, but nevertheless, please try to account for them as much as possible. I would also be happy if you could briefly respond to the most critical comments I had.**

**Please also note that your paper must finish with a conclusions section. Thus, please split the current discussion section into a discussion and a conclusion.**

**I think you should be able to incorporate my suggestions (see attached annotated PDF file) rather quickly.**

**Yours sincerely**

**Dominik Brunner**

We thank the editor for his detailed review of our text and for his comments that helped improve the quality of the text and of the discussions.

**L17-19 It would be good if you could expand the sentence a bit explaining why the classical tracer release method is inferior, e.g. with "… than the tracer release method which does not account for xyz".**
We prefer to add some sentences earlier (see above) that describe the principle and aims of the new concept, including the aim at overcoming the problems of the classic tracer release technique when the sources are not well collocated.

**L24-25 "more complex implementations", not quite clear what you mean.**
The sentence has been changed into "Further studies and more complex implementations with more advanced transport models and more advanced optimizations of their configuration will be required to generalize the applicability of the approach and strengthen its robustness."

**L36 "Many emitting industrial sites have a typical size of 100-500 m²", this would be 10 m x 10 m or 20 m x 25 m. This is way too small.**
We agree with the fact that industrial sites can larger than 500 m², the upper value has been changed into 1km².

**L47-50 "mobile measurements can be conducted", tracers (like SF6) have also be released over much longer times (days to weeks) but then in combination with stationary measurements.**
We prefer to add "generally" and change "and" into "or" in the sentence and avoid referring to experiments with measurements at fixed sites to avoid complex discussions at this stage and keep the logical flow with the next sentences.

**L86-88 "emission spatial distribution", Depends on how the inversion is setup. Probably " prior emissions" would be more correct.**

We feel that applying the two suggested corrections would lead to a wrong assessment of the typical situations that are discussed in this study. We prefer to rephrase the sentence into "However, the skill of such approaches strongly relies on a good accuracy of the transport modelling and on the ability to characterise the statistics of the modelling uncertainties. It can also strongly rely on the prior knowledge on the emissions, in particular on the spatial distribution of the multiple sources within an industrial site for the type of applications considered in this study, and on the ability to characterise the uncertainties in such a knowledge."

**L104-107 "was no strong incentives to report emissions", I disagree. The Kyoto Protocol and probably national legislations provided a strong incentive to report emissions (look at the E-PRTR database for point sources in Europe, for example), but there was not strong incentive to actually measure the emissions from this type of sources.**

The sentence has been changed: "Until recently, there were no strong incentives to estimate site emissions using dedicated measurements. The reported estimates were usually derived using…"

**L156-159 If there is no street within the boundaries of the two criteria mentioned above, then you cannot apply the method, or only by compromising on quality. "Adapt" is a bit misleading here.**

The sentence has been changed into: "Finally, the choice of the distance is constrained by the need for conducting measurements on roads located downwind of the site sources (depending on the specific wind directions during the measurement campaigns) when using instruments onboard cars as in this study."

**L160-162 Quite complicated and unclear sentence. How can a "combination" be "linear"? Probably you are trying to say that the relation between the concentrations and the emission rates is linear and expressed by the observation operator H?**

The sentence has been changed into: "The simulated relation between the gas emission rates from the single or multiple sources of the site and the atmospheric concentrations relies on the knowledge of the location and spread of each source and on the proxy of the atmospheric transport. It is linear and expressed by the observation operator **H**."

**L243-247 I would think that the most commonly used models for this are Lagrangian dispersion models driven by flow and turbulence fields either diagnosed or explicitly computed (with a CFD type solver). Spray, AUSTAL2000 or GRAL are a few examples for such models that are used by many engineering companies in Europe and also certified for use in legislatory applications.**

The sentence has been changed into: "Many types of transport models are used to simulate the dispersion of pollutants at the local scale, i.e. typically over distances from a few metres to 1 or 2 kilometres, from simple Gaussian models to the coupling of Lagrangian and sophisticated CFD

(Computational Fluid Dynamics) models that allow to determine turbulent patterns for complex terrain through an explicit representation of reliefs and obstacles."

**L 322-324 I did not understand this last sentence. I have the feeling that it is merely repeating the statement of the previous sentence and that it can therefore be dropped. I agree with reviewer #1 that your paper is very wordy with unnecessary long sentences stating facts that are already clear from the context. It has much improved, and I hope my suggestions will further help making the paper more concise.**
We have deleted this sentence.

**L350-353 Too complicated and too long. I can only guess what you are trying to say here.**
We have removed this sentence, the underlying idea was too complicated and partly redundant with the previous sentence.

**L459-466 Is this last paragraph really needed? I did not fully understand it.**
This paragraph has been deleted.

**L491-492 30 seconds is not at all "short" but is in fact extremely long for such a mobile system. Probably you used a too large tubing for the small sample flow.**
We agree to this comment and replace the sentence by: "The sampled air was sent into the instrument by an external pump system allowing an inlet lag between the sample inlet and the measurements of less than 30 seconds. This more or less constant inlet lag introduced a spatial offset when comparing the measured and modelled tracer or methane concentrations."

**L514-516 Systematically good does not mean anything. If you have really compared, you should be able to give a number for the differences.**
The relative difference between flow rate calculated with the mass difference and the flow rate read on the flowmeter was between 1 and 3 % depending depending on the series. These numbers have been added to the manuscript.

**L602-604 I found this approach of selecting a specific class a bit weird. Some sort of direct estimation of the dispersion parameters sigma_y and sigma_z (or of the parameters alpha, beta gamma) would have seemed much more logical to me. By selecting a class, you can never fit the measured indices exactly (or only by chance), and will thus always have to deal with a biased simulation. These biases may cancel out with sufficient transects, but I wouldn't take this for granted.**
We have added a discussion on this topic in section 4. Such a direct optimization of these parameters could make sense but should be carefully handled to avoid obtaining inconsistent (inconsistent between them and strongly inconsistent with the meteorological conditions) values for these different parameters to fit mathematically the tracer plume indices. By following the Briggs formulation, we ensure a minimum of physical consistency of our model parameterization.

**L729-732 If I understand correctly, you are defining a different observation uncertainty for each transect. This makes sense to me, but then I don't understand the following statement in the discussions section (line 880):"unlike the estimate of uncertainties for the tracer release technique, the statistical inversion ignores the variations of the methane observation values and methane model data misfits from one transect to the other one"**

Here we speak about defining R based on tracer data, while the sentence from section 4 discusses methane data: it states that the computation of the posterior uncertainties strongly relies on the configuration of R, B and H but not on the values of the methane model – data misfits. We still update the sentence in section 4 since the factor between the measured tracer and methane plume indices enters into the computation of R.

**This discussion section is very wordy and not at all to the point. In this way, the main points of the paper are in danger of getting overlooked. Please note that your paper must finish with a conclusions section. Thus, please split this section into a discussion and a conclusion.**
We have created a conclusion section.

**L806-808 This statement is incompatible with the large biases produced by the tracer release technique in configurations 2 and 4, and actually it is incompatible with the conclusions of the whole paper.**
This statement should be place into a context in which the methane emissions from industrial sites are poorly known and so for which having estimates with 30% uncertainty should already be good. Our correction of the sentence should increase the consistency with the results.

**L883-885 I have to admit that I find the low uncertainties very disturbing and, if true, a major deficiency of the method. The method should not only give a good estimate of the flux but also of its uncertainty. It is difficult to understand how you can get from an initial 80% uncertainty of the prior down to an uncertainty of often less than 1% in the posterior. I also don't think that it is a good idea to use the misfit between simulated and measured acetylene tracer index to define the transport uncertainty, because the transport model could (or I would even say should) be optimized in such a way that it more or less perfectly matches the measured index. In this case, the transport error would go to zero.**
We agree about the fact that the model error, and thus the overall observation error, can be under-estimated since comparing modeled and measured plume indices to derive it, while the model was optimized to fit these indices. By designing such a derivation of R, our assumption is that the main transport error arises from the unability of the selected class of stability to perfectly fit with the tracer plume index. The use of statistics of model - data misfits for tracer plume indices over slices (as in the fourth configuration) rather than over the whole plumes should better reflect the transport errors arising from turbulent, background patterns, and from the inhomogeneity of the wind field. We now discuss it here.